

# Glacial cycles simulation of the Antarctic Ice Sheet with PISM - Part 2: Parameter ensemble analysis

Torsten Albrecht [1], Ricarda Winkelmann [1,2], and Anders Levermann [1,2,3]

[1]Potsdam Institute for Climate Impact Research (PIK), Member of the Leibniz Association, Potsdam, Germany
[2]Institute of Physics and Astronomy, University of Potsdam, Potsdam, Germany
[3]Lamont-Doherty Earth Observatory, Columbia University, New York, USA

*Correspondence to:* T. Albrecht (albrecht@pik-potsdam.de)

**Abstract.** The Parallel Ice Sheet Model (PISM) is applied to the Antarctic Ice Sheet over the last two glacial cycles ($\approx$ 210,000 years) with a resolution of 16 km. A Large Ensemble of 256 model runs is analyzed in which four relevant model parameters have been systematically varied using full-factorial parameter sampling. Parameters and plausible parameter ranges have been identified in

a companion paper (Albrecht et al., 2019) and are associated with ice dynamics, climatic forcing, basal sliding and bed deformation and represent distinct classes of model uncertainties. The model is calibrated against both modern and geologic data, including reconstructed grounding line locations, elevation-age data, ice thickness and surface velocities as well as uplift rates. An aggregated score is computed for each ensemble member that measures the overall model-data misfit, including mea-

surement uncertainty in terms of a Gaussian error model (Briggs and Tarasov, 2013). The statistical method used to analyze the ensemble simulation results follows closely the simple averaging method described in Pollard et al. (2016).

This analysis further constrains relevant model and boundary parameters by revealing clusters of best fit parameter combinations. The ensemble of reconstructed histories of Antarctic Ice Sheet

volumes provides a score-weighted likely range of sea-level contributions since the Last Glacial Maximum of $9.4 \pm 4.1$ m (or $6.5 \pm 2.0 \times 10^6$ km$^3$), which is at the upper range of previous studies. The last deglaciation occurs in all ensemble simulations after around 12,000 years before present, and hence after the Meltwater Pulse-1A. Our Large Ensemble analysis also provides well-defined parametric uncertainty bounds and a probabilistic range of present-day states that can be used for

PISM projections of future sea-level contributions from the Antarctic Ice Sheet.



## 1 Introduction

Sea-level estimates involve high uncertainty in particular with regard to the potential instability of marine-based parts of the Antarctic Ice Sheet (e.g., Weertman, 1974; Mercer, 1978; Slangen et al., 2017). Processed-based models provide the tools to evaluate the currently observed ice sheet changes

(Shepherd et al., 2018a, b), to better distinguish between natural drift/variability and anthropogenic drivers (Jenkins et al., 2018) and to estimate future changes for possible climatic boundary conditions (Oppenheimer and Alley, 2016; Shepherd and Nowicki, 2017; Pattyn, 2018). Regarding the involved variety of uncertain parameters and boundary conditions, confidence of future projections from such models is strengthened by systematic calibration against modern observations and past

reconstructions. We can build on experience gained in several preceding Antarctic modeling studies (Briggs et al., 2013, 2014; Whitehouse et al., 2012a; Golledge et al., 2014; Maris et al., 2014, 2015; Pollard et al., 2016, 2017), providing paleo dataset compilations or improved calibration algorithms. Modern datasets encompass ice thickness, grounding line and calving front position (Fretwell et al., 2013), surface velocity (Rignot et al., 2011) as well as uplift rates from recent GPS measurements

(Whitehouse et al., 2012b). Reconstructions of grounding line location at the Last Glacial Maximum (LGM) as provided by the RAISED Consortium (Bentley et al., 2014, personal communication Stewart Jamieson) were used as paleo constraints as well as grounding line locations and cosmogenic elevation–age data from the AntICEdat database (Briggs and Tarasov, 2013) at specific sites during the deglaciation period.

In this study we run simulations of the entire Antarctic Ice Sheet with the Parallel Ice Sheet Model (PISM, Winkelmann et al., 2011; The PISM authors, 2017). The hybrid of two shallow approximations of the stress balance and the comparably coarse resolution of 16 km allow for running an ensemble of simulations of ice sheet dynamics over the last two (dominant) glacial cycles, each lasting for about 100,000 years (or 100 kyr). The three-dimensional evolution of the enthalpy within the

ice sheet accounts for the formation of temperate ice (Aschwanden and Blatter, 2009; Aschwanden et al., 2012) and for the production of sub-glacial water (Bueler and van Pelt, 2015). We use a non-conserving sub-glacial hydrology model to determine the effective pressure on the saturated till. The so-called till friction angle (accounting for small-scale till strength) and the effective pressure enter the Mohr-Coulomb yield stress criterion (Cuffey and Paterson, 2010) and hence the pseudo plastic

sliding law, which relates basal sliding velocity to basal shear stress.

PISM comes with a computationally-efficient generalization of the Elastic-plate Lithosphere with Relaxing Asthenosphere Earth model (Lingle and Clark, 1985; Bueler et al., 2007) with spatially varying flow in a viscous upper mantle half-space below the elastic plate, which does not require relaxation time as parameter. Geothermal heat flux based on airborne magnetic data from Martos et al.

(2017) is applied. Climate boundary conditions are based on mean precipitation from Racmo2.3p2 (Wessem et al., 2018) and a temperature parameterization based on ERA-Interim re-analysis data (Simmons, 2006) in combination with the empirical Positive-Degree-Day method (PDD, e.g., Reeh,





1991). Climatic forcing is based on ice-core reconstructions from EPICA Dome C (EDC, Jouzel et al., 2007) and WAIS Divide ice core (WDC, Cuffey et al., 2016) as well as on sea-level recon-

structions from the ICE-6G GIA model (Stuhne and Peltier, 2015, 2017). Sub-shelf melting in PISM is calculated via PICO (Reese et al., 2018) from salinity and temperature in the lower ocean layers on the continental shelf (Schmidtko et al., 2014) in 18 separate basins based on (Zwally et al., 2015) adjacent to the ice shelves around the Antarctic continent. A description of PISM for paleo applications and sensitivity of the model to various uncertain parameter and boundary conditions are

discussed in a companion paper (Albrecht et al., 2019).

Here, we explore uncertain model parameter ranges related to ice-internal dynamics and boundary conditions (e.g. climatic forcing, bedrock deformation and basal till properties), and use the Large Ensemble approach with full-factorial sampling for the statistical analysis, following Pollard et al. (2016). This procedure yields an aggregated score for each of the 256 ensemble simulations, which

measures the misfit between PISM simulation and 9 equally weighted types of datasets. Each score can be associated with a probabilistic weight to compute the average envelope of simulated Antarctic Ice Sheet and equivalent sea-level histories and hence providing data-constrained present-day states that can be used for projections with PISM.

## 2 Ensemble analysis

Ice sheet model simulations generally imply uncertainties in used parameterizations and applied boundary conditions. In order to generate uncertainty estimates for reconstructions of the Antarctic Ice Sheet history and equivalent sea-level envelopes we employ an ensemble analysis approach that uses full-factorial sampling, i.e., one run for every possible combination of parameter values. We

follow here closely the simple-averaging approach used in Pollard et al. (2016). This method yields as reasonable results for an adequately resolved parameter space as more advanced statistical techniques with means of interpolating results between sparsely separated points in multi-dimensional parameter space.

### 2.1 Ensemble parameter

We have identified four relevant independent PISM ensemble parameters with a prior range for each parameter capturing different uncertainties in ice flow dynamics, glacial climate, basal friction and bedrock deformation. The four parameters and the four values used in the ensemble analysis are:

ESIA: Ice-flow enhancement parameter of the stress balance in Shallow Ice Approximation (SIA; Morland and Johnson, 1980; Winkelmann et al., 2011, Eq. 7). Ice deforms more easily in

shear for increasing values of 1, 2, 4 and 7 (non-dimensional) within the Glen-Paterson-Budd-Lliboutry-Duval law connecting strain rates $\dot{\epsilon}$ and deviatoric stresses $\tau$ for ice softness $A$,





which depends on both liquid water fraction $\omega$ and temperature $T$ (Aschwanden et al., 2012),

$$\dot{\epsilon}_{ij} = \mathrm{ESIA} \cdot A(T,\omega)\, \tau^{n-1}\, \tau_{i,j} \tag{1}$$

PPQ: Exponent $q$ used in "pseudo plastic" sliding law which relates bed-parallel basal shear stress
$\boldsymbol{\tau}_b$ to sliding velocity $\boldsymbol{u}_b$ in the form

$$\boldsymbol{\tau}_b = -\tau_c \frac{\boldsymbol{u}_b}{u_0^{\,q}\, |\boldsymbol{u}_b|^{1-q}}, \tag{2}$$

as calculated from the Shallow Shelf Approximation (SSA) of the stress balance (Bueler and
Brown, 2009), for threshold speed $u_0$ and yields stress $\tau_c$. The sliding exponent hence covers
uncertainties in basal friction. Values are 0.25, 0.5, 0.75 and 1.0 (non-dimensional).

PREC: Precipitation scaling factor $f_p$ according to temperature forcing $\Delta T$ motivated by Clausius-
Clapeyron-relationship and data analysis (Frieler et al., 2015), which can be formulated as
exponential function (Ritz et al., 1996; Quiquet et al., 2012) as

$$P(t) = P_0 \exp\left(f_p\, \Delta T(t)\right) \approx P_0 \left(1.0 + f_p\, \Delta T(t)\right). \tag{3}$$

For given present-day mean precipitation field $P_0$, the factor $f_p$ captures uncertainty in cli-
matic mass balance, in particular for glacial periods. Values are 2, 5, 7 and 10 %/K.

VISC: Mantle viscosity determines the characteristic response time of the linearly viscous half-
space of the Earth (overlain by an elastic plate lithosphere) to changing ice and adjacent ocean
loads (Bueler et al., 2007, Eq. 1). It covers uncertainties within the Earth model for values of
$1 \times 10^{19}$, $5 \times 10^{19}$, $25 \times 10^{19}$ and $100 \times 10^{19}$ Pa s.

## 2.2 Misfit evaluation with respect to individual data-types

With four varied parameters with each parameter taking four values the ensemble requires 256 runs.
For an easier comparison to previous model studies, results are analyzed using the simple averaging
method (Pollard et al., 2016). It calculates an objective aggregate score for each ensemble member
that measures the misfit of the model result to a suite of selected observational modern and geologic
data. The inferred misfit score is based on a generic form of an observational error model assuming
a Gaussian error distribution with respect to any observation interpretation uncertainty (Briggs and
Tarasov, 2013, Eq. 1).

Present-day ice sheet geometry (thickness and grounding line position) provide the strongest spa-
tial constraint of all data-types and also offer a temporal constraint in the late Holocene. Gridded
datasets are remapped to 16 km model resolution. Most of the present-day observational constraints
follow closely the definitions in Pollard et al. (2016, Appendix B, Approach (A)), but weighted with
each grid-cell's specific area with respect to stereographic projection. We added observed modern
surface velocity as additional constraint and expanded the analysis to the entire Antarctic Ice Sheet.



1. TOTE: Mean-square error mismatch of present-day *grounded areas* to observations (Fretwell et al., 2013) assuming uncertainty in grounding line location of 30 km, as in Pollard et al. (2016, Appendix B1). Mismatch is calculated relative to the continental domain, that is defined here as area with bed elevation above -2500 m.

2. TOTI: Mean-square error mismatch of present-day floating *ice shelf areas* to observations (Fretwell et al., 2013) assuming uncertainty in grounding line and calving front location of 30 km, according to Pollard et al. (2016, Appendix B2).

3. TOTDH: Mean-square-error model misfit of present-day state to observed *ice thickness* (Fretwell et al., 2013) with respect to an assumed observational uncertainty of 10 m and evaluated over the contemporary grounded region, close to Pollard et al. (2016, Appendix B3).

4. TOTGL: Mean-square-error misfit to observed *grounding line location* for the modeled Antarctic grounded mask (ice rises excluded) using a signed distance[1] field. This method is different to the GL2D constraint used in Pollard et al. (2016, Appendix B5), and is only applied to the present-day grounding line around the whole Antarctic Ice Sheet according to Fretwell et al. (Bedmap2; 2013) and considering observational uncertainty of 30 km as in TOTI and TOTE above.

5. UPL: Mean-square-error model misfit to modern GPS-based uplift rates on rock outcrops at 35 individual sites using the compilation by Whitehouse et al. (2012b, Table 2) including individual observational uncertainty. Misfit is evaluated for the closest model grid point as in Pollard et al. (2016, Appendix B8), including intra-data type weighting (Briggs and Tarasov, 2013, Sect. 4.3.1).

6. TOTVEL: Mean-square error misfit in (grounded) surface ice speed compared to a remapped version of observational data by Rignot et al. (2011) including their provided grid-cell wise standard deviation, bounded below by 1.5 m/yr.

Paleo-data type constraints are partly based on the AntICEdat compilation by Briggs and Tarasov (2013, Sect. 4.2), following closely their model-data misfit computation. This compilation also includes records of regional sea-level change (RSL), which has not been considered in this study since most of the sea-level signal is a result of the sea-level forcing with up to 140 m rather than the model's ice dynamical response expressed in terms of sea-level equivalents, as PISM lacks a self-consistent sea-level model. According in Pollard et al. (2016, Appendix B4) we evaluate past and present grounding line location along four relevant ice shelf basins.

7. TROUGH: Mean-square error misfit of modeled grounding line position along four transects through Ross, Weddell and Amery Basin and Pine Island Glacier at the Last Glacial Maximum

---

[1]https://pythonhosted.org/scikit-fmm





(20 kyr BP) as compared to reconstructions by Bentley et al. (RAISED Consortium 2014, Scenario A) and at present day compared to Fretwell et al. (Bedmap2 2013), both remapped to model grid. An uncertainty of 30 km in the location of the grounding line is assumed as in Pollard et al. (2016, Appendix B4), but as mean over those two most confident dates and for all four mentioned troughs. In contrast to previous model calibrations, reconstructions of the grounding line position at 15, 10 and 5 kyr BP have not been taken into account here, as they would favor simulations which reveal a rather slow and progressive grounding line retreat through the Holocene in both Ross and Ronne Ice Shelf, which has not necessarily the case (Kingslake et al., 2018).

8. ELEV: Mean of squared misfit of past (cosmogenic) surface elevation vs. age in the last 120 kyr based on model–data differences at 106 individual sites (distributed over 26 regions, weighted by inverse areal density, see Sect. 4.3.1 in Briggs and Tarasov (2013)). For each data-point the smallest misfit to observations is computed for all past ice surface elevations (sampled every 1 kyr) of the 16 km model grid interpolated to the core location and datum as part of a thinning trend (Briggs and Tarasov, 2013, Sect. 4.2). A subset of these data has been also used in Maris et al. (2015); Pollard et al. (2016, Appendix B7).

9. EXT: Mean of squared misfit of observed ice extent at 27 locations around the AIS in the last 28 kyr with dates for the onset of open marine conditions (OMC) or grounding line retreat (GLR). The modeled age is computed as the most recent transition from grounded to floating ice conditions considering the sea-level anomaly. The 1 kyr model output is interpolated down to core location and linearly interpolated to 100 yr temporal resolution, while weighting is not necessary here, as described in Briggs and Tarasov (2013, Sect. 4.2). A subset of these data has been also used in Maris et al. (2015)

## 2.3 Score aggregation

Each of the misfits above are first transformed into a normalized individual score for each data type $i$ and each run $j$ using the median over all misfits $M_{i,j}$ for the 256 simulation. The procedure closely follows Approach (A) in Pollard et al. (2016, Sect. 2.4.1). Then the individual score $S_{i,j}$ is normalized according to the median to

$$S_{i,j} = \exp\left(-M_{i,j} / \operatorname{median}(M_{i,j=1..256})\right). \tag{4}$$

As in Pollard et al. (2016) we also assume that each data type is of equal importance to the overall score, avoiding the inter-data-type weighting used by Briggs and Tarasov (2013); Briggs et al. (2014), which would favor data types of higher spatio-temporal density. Hence the aggregated score for each run $j$ is the product of the nine data-type specific scores, according to the score definition





in Approach (A) by Pollard et al. (2016)

$$S_j = \prod_{i=1..9} S_{i,j}.$$  (5)

This implies, that one simulation with perfect fit to eight data types, but one low individual score, yields a low aggregated score for this simulation and hence for instance a low confidence for future applications.

## 3   Results

### 3.1   Analysis of parameter ensemble

We have run PISM simulation over the last glacial cycle. Figure 1 shows the aggregate scores $S_j$ for each of the 256 ensemble members, over the 4-D space spanned by the parameters ESIA, PPQ, PREC and VISC. Each individual sub-panel shows PPQ vs. VISC, and the sub-panels are arranged

left-to-right for varying PREC and bottom-to-top for varying ESIA. Scores are normalized by the best score member, which equals value 1 here.



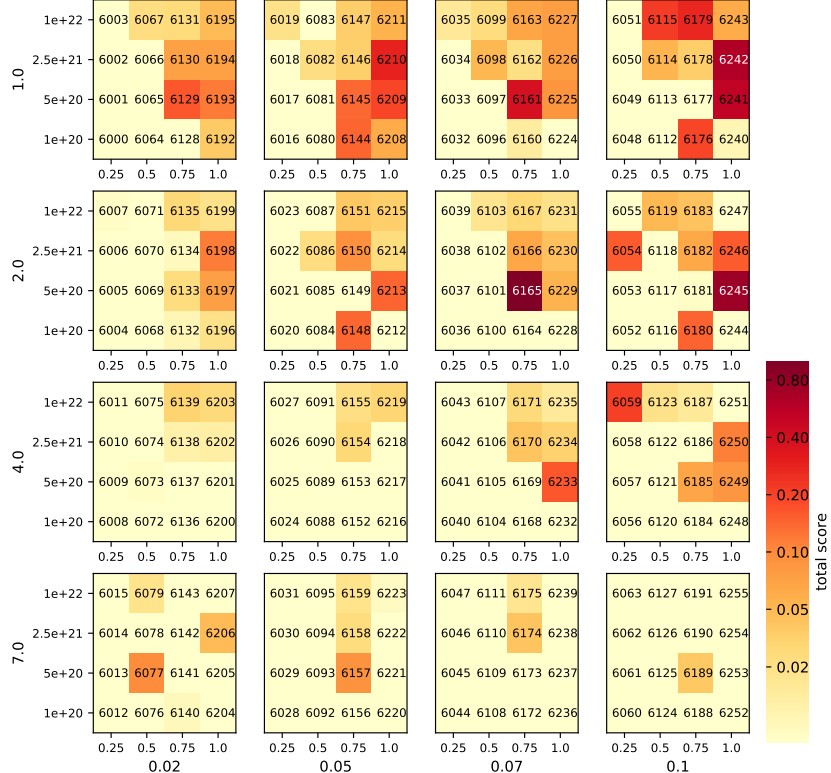

Figure 1: Aggregated score for all 256 ensemble members (4 model parameters, 4 values each) showing the distribution of the scores over the full range of plausible parameter values. The score values are computed versus geologic and modern data sets, normalized by the best score in the ensemble, and range from <0.01 (bright yellow, no skill) to 1 (dark red, best score) (cfs. Pollard et al., 2016, Figs. 2 + C1), on a logarithmic color scale. The four parameters are the SIA enhancement factor ESIA (outer y-axis), the temperature-dependent precipitation scaling PREC (outer x-axis), the mantle viscosity VISC (inner y-axis) and the power-law sliding exponent PPQ (inner x-axis).

The parameter ESIA enhances the shear-dominated ice flow and hence ice thickness particularly in the interior of the ice sheet and therewith the total ice volume. ESIA values of 4 or 7 have been used in other models (e.g., Maris et al., 2015) to compensate for underestimated ice thickness in the interior of East Antarctic Ice Sheet under present-day climate conditions. In our ensemble we find a trend towards higher scores for small ESIA with values of 1 or 2 (in the upper half section of Fig. 1). This becomes more prominent when considering ensemble-mean score shares for individual parameter values as in Fig. 3, with a normalized mean score of 46% for ESIA = 1 as compared to a mean score of 6% for ESIA = 7. Most of this trend is a result of the individual data-type score TOTDH (see





Fig. 5, column 4, row 3) as it measures the overall misfit of modern ice thickness (and volume distri-
bution). Partly this trend can be also attributed to the TROUGH data-type scores (Fig. 5, column 8,
row 3), as for higher ESIA values grounding line motion is slowed down and the time until present
is not sufficient for a complete retreat back to the observed present-day location, at least in some
ice shelf basins. The best score ensemble members for small ESIA values are found in combination
with high values of mantle viscosity VISC and high values of friction exponent PPQ (center column
panels in Fig. 4).

Regarding the choice of the precipitation scaling PREC the best-score members are found at the
upper range with values of 7 %/K or 10 %/K (see right half section in Fig. 1). Considering normal-
ized ensemble-mean score for individual parameter values over the full range of 2–10 %/K we can
find a trend from 13% to 42% (see lowest panel in Fig. 3). Regarding combinations of parameter with
PREC (left-hand column in Fig. 4), we detect a weak trend towards lower ESIA and higher PPQ,
while individual data-type scores (lower row in Fig. 5) show a rather trendless patterns, in particular
regarding the misfit to present-day obsrvations. As the PREC parameter is linked to the temperature
anomaly forcing, it affects the ice volume and hence the grounding line location particularly for tem-
perature conditions different from present day. This suggests a stronger signal of PREC parameter
variation in the paleo data-types scores.

A more complex pattern is found for PPQ in each of the sub-panels of Fig. 1 with highest scores for
values of 0.75 and 1.0. Averaged over the ensemble and normalized over the four parameter choices
we find mean scores of 5% for PPQ = 0.25 (and hence rather plastic sliding) while best scores are
found for PPQ = 0.75 and PPQ = 1.0 (linear sliding) with mean scores of 40% and 46%, respectively
(see second panel in Fig. 3). The mean score reveal best scores in combination with medium mantle
viscosity VISC between $0.5 \times 10^{21}$ Pa s and $2.5 \times 10^{21}$ Pa s, as visible in the upper right panel of
Fig. 4. As sliding mainly affects the ice stream flux, the score trend over the range of PPQ values
mainly results from velocity misfit data-type TOTVEL and grounding line position related data-
types (TOTE, TOTGL and THROUGH), see Fig. 3 (second row).

Regarding mantle viscosity VISC, scores are generally low with 9% for the smallest sampled value
of the parameter VISC = $10^{20}$ Pa s, while best scores are found in the ensemble for the five times
larger viscosity of VISC = $5 \times 10^{20}$ Pa s with 44%. This value is also used in the GIA model ICE-6G
(Peltier et al., 2015), but in some localized regions as in the Amundsen Sea, upper mantle viscosities
could be even smaller up to the order of $10^{19}$ Pa s (Barletta et al., 2018). For the upper range of
mantle viscosities up to VISC = $10^{22}$ Pa s we find a normalized ensemble mean of 27% and 20%,
respectively. Note that VISC parameter values have been sampled non-linearly over a range of two
orders of magnitude. The trend stems mainly from the misfit of present-day uplift rates expressed as





data-type score TOTUPL. For the lowest value there is a clear trend towards smaller scores in the grounding-line and ice-thickness related data-types, such as TOTE, TOTGL, TROUGH and TOTDH respectively. As mantle viscosity determines the rate of response of the bed to changes in ice thick-

ness a low viscosity corresponds to a rather quick uplift after grounding line retreat and hence to a retarded retreat and hence to a rather extended present-day state. In contrast, rather high mantle viscosities involve a slow bed uplift and grounding line retreat can occur faster. More specifically, in the partially over-deepened ice shelf basins, which have been additionally depressed at the Last Glacial Maximum by a couple of hundred meters as compared to present, grounding line retreat

can amplify itself in terms of a regional Marine Ice Sheet Instability (Mercer, 1978; Schoof, 2007; Bart et al., 2016). The best score ensemble members are found for intermediate mantle viscosities of VISC=5×$10^{20}$ Pa s and VISC=25×$10^{20}$ Pa s.

The five best-score ensemble members and associated parameter combinations are listed in Ta-

ble 1. The individual scores with respect to the nine data-types are visualized for the best 20 ensemble members in Fig. 2. The scores associated with the paleo data-types ELEV and EXT show only comparably little variation among the ensemble (in total around 0.4). This also applies for the present-day ice shelf area mismatch TOTI (total spread of 0.2), as no calving parameter have been varied. In contrast, present-day data types associated with velocity (TOTVEL) and uplift rates (TO-

TUPL) show strong variations among the best 20 ensemble members , with a spread in score across the for entire ensemble of 0.7 and 0.85 respectively. For data types that are related to grounding line position (TOTGL, TOTE) and ice volume (TOTDH) we find a similar order as for the TOTAL aggregated score, with individual spread in scores of 0.5-0.6 across all ensemble members. Comparing the ensemble-mean present-day ice thickness with observations (Bedmap2; Fretwell et al.,

2013) we find regions in the inner East Antarctic Ice Sheet and in parts of the Weddell Sea sector that are about 200 m too thin, while ice thickness is overestimated by more than 500 m in the Siple Coast, in the Amery basin and along the coast line, where smaller ice shelves tend to be grounded in the simulations (Fig. 6a). Ross Sea, Weddell Sea and Amery basins show the largest ensemble-score weighted standard deviation with more than 500 m ice thickness (Fig. 6b). The ensemble spread in

those basins can be associated with uncertainties in grounding line position, as grounding line has to retreat in time from its extended position at Last Glacial Maximum crossing up to 1000 km long basins, leaving behind the large floating ice shelves (Fig. 7). In about 10% of the score-weighted simulations grounding line remains at the extended position without significant retreat, linked to high basal friction (PPQ=0.25) and an efficient negative feedback on grounding line motion related

to a fast responding bed (low VISC). In contrast, for rather low friction and high mantle viscosities we find fast grounding line retreat, with a stabilization of grounding line position at our even inland of the observed location in 50% or 75% of the score-weighted simulations in the Ross and Weddell Sea sector, respectively (Fig. 8, upper panels). Due to the unloading of the large ice shelf basins the




sea floor lifts up by a few hundred meters which leads to grounding line re-advance supported by the
formation of ice rises (Kingslake et al., 2018). The re-advance in the ensemble mean is up to 100 km,
while some of the best-score simulations reveal temporary ungrounding through the Holocene up to
400 km upstream of the present-day grounding line in the Ross sector. The Amundsen Sea sector
and Amery Ice Shelf do not show such rebound effects in our model ensemble (Fig. 8, lower panels)

| Simulation No. | ESIA | PPQ | PREC | VISC | Score | Normal. Score |
|---|---|---|---|---|---|---|
| 6165 | 2.0 | 0.75 | 7 %/K | $5 \times 10^{20}$ Pa s | $6.1 \times 10^{-3}$ | 1.0 |
| 6245 | 2.0 | 1.0 | 10 %/K | $5 \times 10^{20}$ Pa s | $4.6 \times 10^{-3}$ | 0.76 |
| 6242 | 1.0 | 1.0 | 10 %/K | $25 \times 10^{20}$ Pa s | $3.9 \times 10^{-3}$ | 0.63 |
| 6241 | 1.0 | 1.0 | 10 %/K | $5 \times 10^{20}$ Pa s | $3.2 \times 10^{-3}$ | 0.53 |
| 6261 | 1.0 | 0.75 | 7 %/K | $5 \times 10^{20}$ Pa s | $2.4 \times 10^{-3}$ | 0.39 |

Table 1: Five best-score ensemble parameter combinations with parameter values and total scores.





| | TOTAL | TOTE | TOTI | TOTDH | TOTVEL | TOTGL | TOTUPL | TROUGH | ELEV | EXT |
|---|---|---|---|---|---|---|---|---|---|---|
| 6165 | 1.0 | 0.628 | 0.426 | 0.702 | 0.659 | 0.649 | 0.707 | 0.848 | 0.291 | 0.437 |
| 6245 | 0.756 | 0.622 | 0.482 | 0.662 | 0.691 | 0.646 | 0.691 | 0.824 | 0.273 | 0.336 |
| 6242 | 0.631 | 0.591 | 0.412 | 0.743 | 0.445 | 0.609 | 0.578 | 0.859 | 0.412 | 0.385 |
| 6241 | 0.527 | 0.537 | 0.394 | 0.657 | 0.611 | 0.547 | 0.792 | 0.724 | 0.328 | 0.368 |
| 6161 | 0.392 | 0.529 | 0.399 | 0.651 | 0.663 | 0.565 | 0.66 | 0.703 | 0.271 | 0.371 |
| 6210 | 0.29 | 0.55 | 0.382 | 0.747 | 0.412 | 0.55 | 0.326 | 0.837 | 0.428 | 0.427 |
| 6179 | 0.277 | 0.531 | 0.393 | 0.662 | 0.412 | 0.453 | 0.827 | 0.434 | 0.44 | 0.417 |
| 6115 | 0.218 | 0.539 | 0.407 | 0.642 | 0.28 | 0.452 | 0.809 | 0.464 | 0.428 | 0.467 |
| 6059 | 0.205 | 0.485 | 0.468 | 0.472 | 0.369 | 0.57 | 0.815 | 0.41 | 0.509 | 0.327 |
| 6209 | 0.197 | 0.575 | 0.427 | 0.69 | 0.683 | 0.584 | 0.129 | 0.8 | 0.447 | 0.387 |
| 6176 | 0.196 | 0.488 | 0.335 | 0.586 | 0.633 | 0.425 | 0.612 | 0.682 | 0.321 | 0.348 |
| 6129 | 0.165 | 0.529 | 0.375 | 0.643 | 0.664 | 0.567 | 0.268 | 0.669 | 0.344 | 0.341 |
| 6233 | 0.164 | 0.431 | 0.368 | 0.418 | 0.683 | 0.469 | 0.806 | 0.398 | 0.389 | 0.38 |
| 6246 | 0.158 | 0.558 | 0.379 | 0.632 | 0.351 | 0.609 | 0.363 | 0.799 | 0.349 | 0.334 |
| 6054 | 0.157 | 0.532 | 0.435 | 0.634 | 0.495 | 0.575 | 0.316 | 0.491 | 0.475 | 0.313 |
| 6180 | 0.148 | 0.442 | 0.339 | 0.578 | 0.627 | 0.429 | 0.725 | 0.619 | 0.33 | 0.263 |
| 6144 | 0.145 | 0.473 | 0.348 | 0.572 | 0.614 | 0.48 | 0.307 | 0.682 | 0.402 | 0.38 |
| 6213 | 0.144 | 0.51 | 0.356 | 0.617 | 0.608 | 0.569 | 0.313 | 0.687 | 0.238 | 0.444 |
| 6148 | 0.141 | 0.446 | 0.326 | 0.56 | 0.64 | 0.479 | 0.414 | 0.598 | 0.328 | 0.425 |
| 6145 | 0.14 | 0.668 | 0.508 | 0.729 | 0.645 | 0.671 | 0.057 | 0.881 | 0.351 | 0.452 |

Figure 2: Nine individual scores for 20 (total) best ensemble members computed versus modern and geologic data sets, divided by dashed line. The score values are normalized by the median misfit, and range from 0 (bright yellow, no skill) to 1 (dark red, best score) on a linear color scale. The ensemble spread of some individual paleo data types ELEV and EXT, as well as for present-day ice shelf mismatch TOTI, is comparably low with around 0.4 and 0.2 respectively. In contrast, grounding line location at LGM and present-day along four ice shelf basins (TROUGH) and present-day uplift rates (TOTUPL) have strongest impacts on the aggregated score with more than 0.85 spread in individual scores across the ensemble. Intermediate spread of individual scores show TOTGL with around 0.5, TOTE and TOTDH with around 0.6 and TOTVEL with around 0.7.



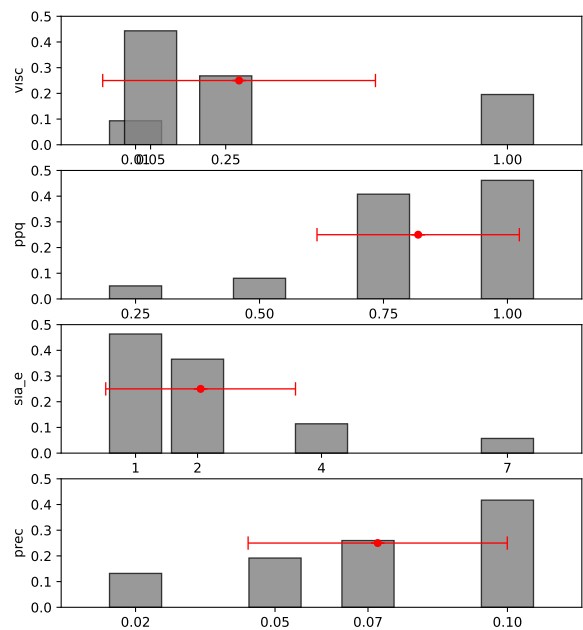

Figure 3: Ensemble-mean scores for individual parameter values (normalized such that sum is 1, or 100%). The weighted mean over the four ensemble-mean scores with standard deviation is shown in red (compare Figs. 3 + C2 in Pollard et al. (2016)).




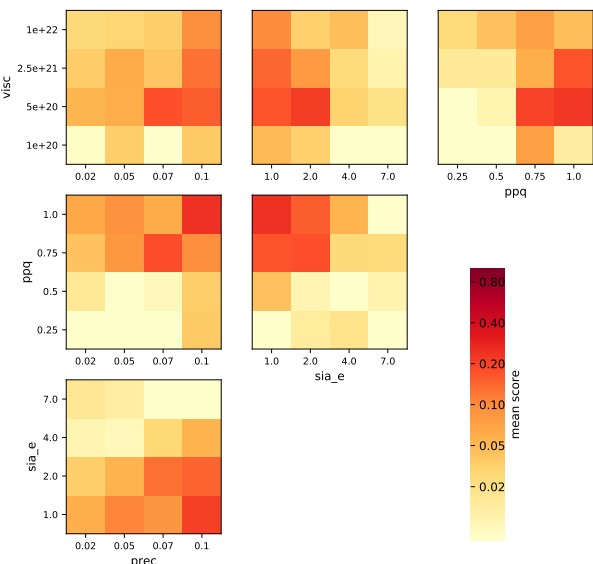

Figure 4: Ensemble-mean scores for six possible pairs of parameter values to visualize parameter dependency (compare Figs. 4 + C3 in Pollard et al. (2016)). Values are normalized such that the sum for each pair is 1. Color scale is logarithmic ranging from 0.01 (bright yellow) to 1 (red).

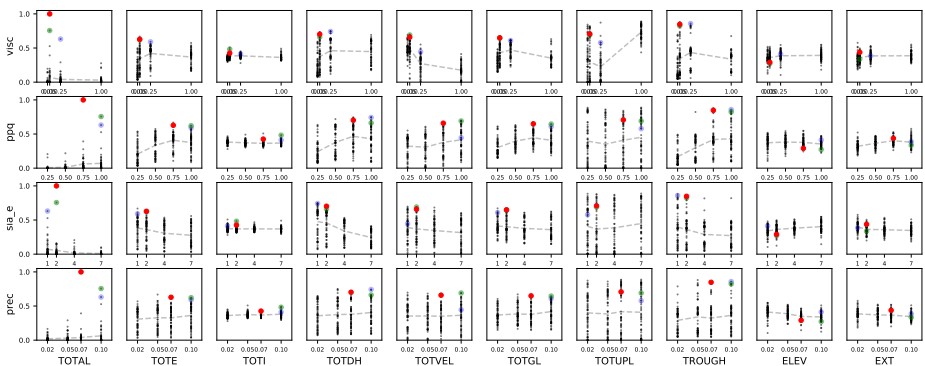

Figure 5: Scatter plot of both aggregated score and the nine individual data-type scores (panels from left to right) for each parameter setting (VISC, PPQ, ESIA and PREC as y-axis). Red dots indicate the best-score member, green and blue the second and third best ensemble members (see Table 1). Grey-dashed line indicates mean score tendency over sampled parameter range.





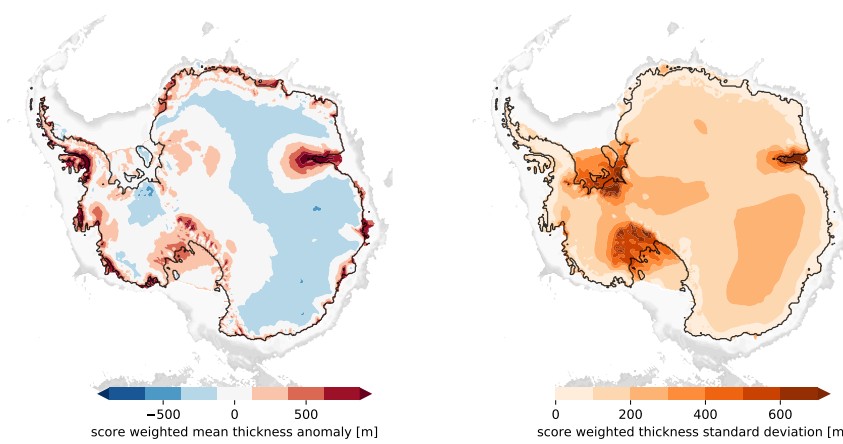

Figure 6: Score-weighted mean ice thickness anomaly to Bedmap2 (left) and score-weighted standard deviation of ice thickness (right). Ice thickness in coastal regions in West Antarctica but also in the Amery basin are generally overestimated. Amery and Filchner-Ronne Ice Shelves and Siple Coast region reveal the highest standard deviation in reconstructed present-day ice thickness among the ensemble members.





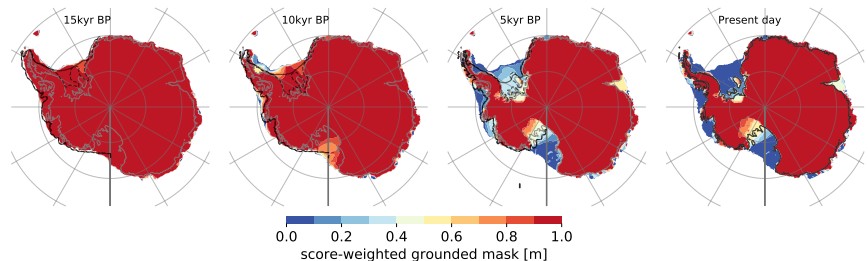

Figure 7: Ensemble-score weighted grounded mask for 5 kyr snapshots. Mask value 1 (red) indicates grounded area which is covered by all simulations, while blueish colors indicate areas which are covered only by a few simulations with low scores (compare Fig. D4 in Pollard et al. (2016)). For the last two snapshots, grounding line in the Ross Sea and Weddell Sea sector is found in about 50% of score-weighted simulations inland of its present location (Fretwell et al., 2013, grey line) with some grounding line re-advance (Kingslake et al., 2018). In contrast less than 10% show no grounding line retreat from glacial maximum extent. Black lines indicate reconstructions by the RAISED Consortium (Bentley et al., 2014, Scenario B solid and scenario A dashed).



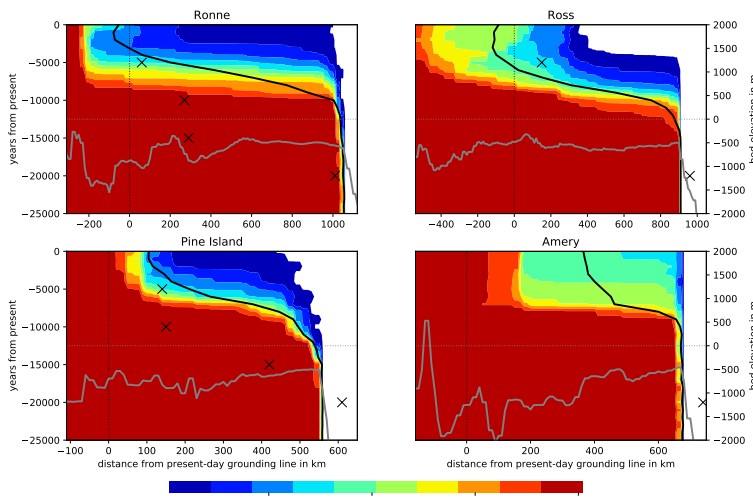

Figure 8: Ensemble score-weighted grounded ice cover along transects trough Weddell, Ross, Amundsen and Amery Ice Shelf basins over the last 25 kyr simulation period (left y-axis, compare Fig. D5 in Pollard et al. (2016)). Grounded areas which are covered by all simulations are indicated by value 1 (red), while blueish colors indicate areas which are covered only by some simulations (or those with low scores). Grounding line in the Ross Sea and Weddell Sea sector is found inland of its present location (vertically dotted) within the last 10 kyr simulation time in about 50% and 75% of score-weighted simulations, respectively. The score-weighted mean curve (black) reveals re-advance of the grounding line of up to 100 km as discussed in (Kingslake et al., 2018). Such behavior is not found in the Pine Island trough, where grounding line retreat stops in 90% of simulations at about 200 km downstream of its present day location. Similar in the Amery Ice Shelf, where in 30% of score-weighted simulations the ice shelf does not retreat at all from its LGM extent. Bed topography (Bedmap2; Fretwell et al., 2013) along the transect is indicated as gray line with respect to the right y-axis.

## 3.2 Reconstructed sea-level histories

For the parameter ensemble analysis we have first run four simulations starting in the penultimate interglacial 210 kyr BP. These four simulations use four different values of mantle viscosity covering two orders of magnitude (VISC $= 10^{20} - 10^{22}$ Pa s). They show quite a consistent maximum ice volume at the penultimate glaciation around 130 kyr BP (see violet lines in Fig. 9). Due to the different Earth response times associated with varied mantle viscosities, the curves branch out when the ice sheet retreats. Those four simulations were used as initial states at 125 kyr BP for the other 252 simulations of the large ensemble. At the end of the Last Interglacial (Eemian) at around 120 kyr BP,





when the full ensemble has been run for only 5 kyr, the ensemble mean ice volume is 1.0 m SLE below modern with a score-weighted standard deviation of around 2.7 m SLE (volume of grounded ice above flotation in terms of global mean sea level equivalent as defined in Albrecht et al. (2019). This corresponds to a grounded ice volume anomaly in relation to present day observations of $-0.3 \pm 1.4 \times 10^6$ km$^3$. These numbers may not reveal the full ensemble spread as simulations still carry some memory of the previous glacial cycle simulations with different parameters. On average, grounding lines and calving fronts retreat much further inland at LIG than for present-day conditions. Yet, complete collapse of WAIS does not occur in any of the ensemble members, most likely as a result of intermediate till friction angles and hence higher basal shear stress underneath the inner WAIS (see optimization in Albrecht et al. (2019)). Assuming, that the memory of the previous spin-up has vanished at the Last Glacial Maximum (in our simulations at around 15 kyr BP), where the model ensemble yields a (grounded) Antarctic Ice Sheet volume of $9.4 \pm 4.1$ m above present-day observations, or $6.5 \pm 2.0 \times 10^6$ km$^3$. The histogram of score-weighted sea-level anomalies of all simulations at Last Glacial Maximum actually reveals four distinct maxima at around 4.5, 8.1, 9.0 and 13.0 m SLE (Fig. 10 a), which can be attributed to the five best-score simulations in Table 1. The ensemble spread is hence relatively wide, but still quite symmetric, as comparison with the Gaussian distribution reveals. The LGM ice volume increases for lower PPQ, lower PREC and lower ESIA values, while it seems to be rather insensitive to the choice of VISC. When comparing simulated volumes at Last Glacial Maximum to modeled present-day volumes (such that model biases cancel out) the model ensemble yields $10.0 \pm 4.1$ m of global mean sea level equivalents, or $5.8 \pm 2.0 \times 10^6$ km$^3$.

Most of the retreat from LGM extent and hence most of Antarctica's sea-level rise contribution occurs in our simulations after 10 kyr BP (cf. Fig. 10 b, c). In particular, for higher mantle viscosities we find episodic self-amplified retreat with more than 0.5 cm SLE per year change rate in West Antarctic basins (as in the best-fit simulation at 7.5 kyr BP). This leads in some cases to grounding line migration even upstream of its present location and subsequent re-advance during Holocene due to uplift of the Earth (discussed in Kingslake et al., 2018). However, these rapid episodes of retreat occur in our simulations consistently after Meltwater pulse 1A (MWP1a) around 14.5 kyr (dashed line in Fig. 9). This delay supports the idea, that Antarctic Ice Sheet retreat has been not a source but rather a consequence of the relatively quick rise in global mean sea-level by about 15 m within 350 yr or $\approx 4$ cm/yr (Liu et al., 2016), while core analysis of iceberg-rafted debris suggest earlier and stronger recession of the Antarctic Ice Sheet at MWP1a (Weber et al., 2014). As MWP1a initiated the Antarctic Cold Reversal (ACR) with about two millenia of colder surface temperatures, a freshening of surface waters leading to a weakening of Southern Ocean overturning, resulting in reduced Antarctic Bottom Water formation, enhanced stratification and sea-ice expansion. This could have caused an increased delivery of relatively warm Circumpolar Deep Water onto the continental shelf close to the grounding line and hence to stronger sub-surface melt (Golledge et al., 2014; Fogwill et al., 2017). As our sub-shelf melting module is forced with a modified surface temperature



anomaly forcing, PICO responds with less melt during ACR period and hence prevents from ice
sheet retreat (see also Albrecht et al. (2019) for sensitivity analysis). Also MWP1b around 11.3 kyr
BP occurred well before deglacial retreat initiated in most simulations of our model ensemble (see
Fig.9 c), in contrast to a previous PISM study (Golledge et al., 2014). One key parameter for the
onset of retreat could be the minimal till friction angle on the continental shelf with values possibly

below 1.0°, see Appendix A The timing of deglaciation and possible rebound effects can explain a
natural drift in certain regions that lasts until present-day. In the score-weighted average the ensemble
simulations suggest a sea-level contributions over the last 3,000 model years of about 0.25, mm/yr,
while for the reference simulation the Antarctic ice above flotation is even slightly growing (cf.
Fig. 9 c), partly explained by net uplift in grounded areas (Fig. 12).




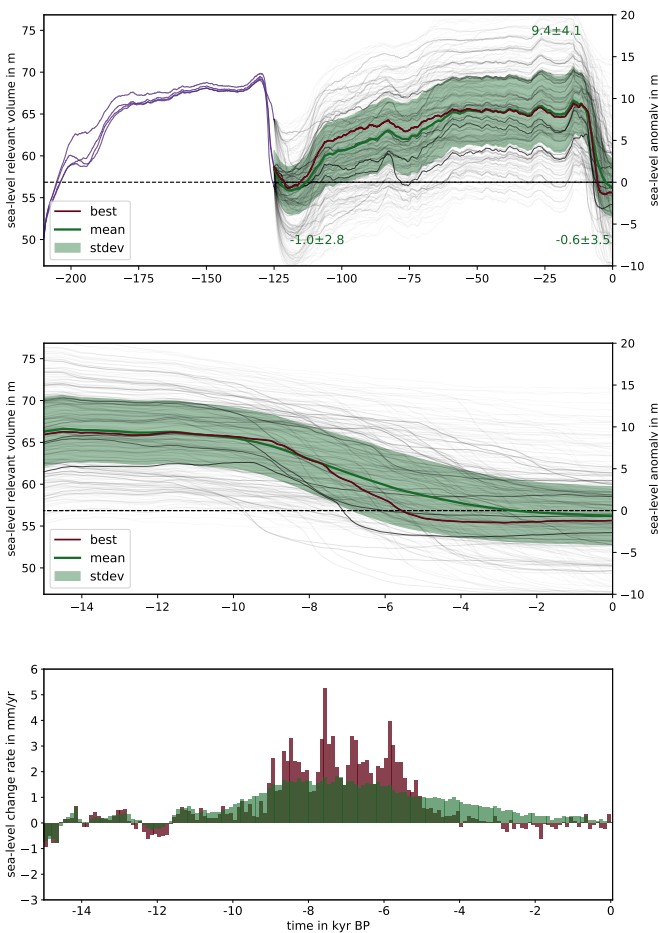

Figure 9: Simulated sea-level relevant ice volume histories over the last glacial cycle(s) (upper) and for last deglaciation (middle) for all 256 individual runs in the parameter ensemble, transparency weighted by aggregated score. Red line indicates the best-score run, the green line indicates the score-weighted ensemble mean and standard deviation. At Last Glacial Maximum (here at 15 kyr BP) the reconstructed ensemble-mean ice volume above flotation yields 9.4±4.1 m SLE above present-day observation (compare to Figs. 5 + C4 in Pollard et al. (2016)). Violet lines indicate simulations over the penultimate glacial cycle with four different mantle viscosities, from which the large ensemble branches at 125 kyr BP. During deglaciation the score-weighted ensemble mean (green) shows most of the sea-level change rates (lower panels) between 9 kyr BP and 5 kyr BP with mean rates above 1 mm yr$^{-1}$, while the best-score simulation (red) reveals rates of sea-level rise of up to 5 mm yr$^{-1}$ (100 yr bins) in the same period (cf. Golledge et al., 2014, Fig. 3 d). In contrast to the ensemble mean, the best-score member (red line) shows a minimum ice volume in the mid-Holocene (around 4 kyr BP) and subsequent regrowth.





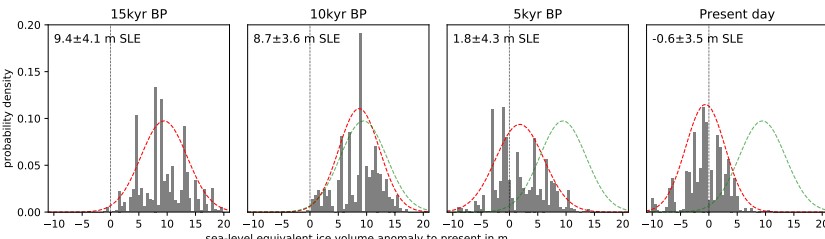

Figure 10: Equivalent global-mean sea-level contribution (ESL) relative to modern at every 5 kyr over the last deglaciation period. Grey bars show the score-weighted ensemble distribution (0.5 m bins), the red curve indicates the statistically likely range (normal distribution) of the simulated ice volumes with width of 1-sigma standard deviation as for the green envelope in Fig. 9, green gaussian curve from 15k kyr snapshot for comparison (compare to Figs. 6 + C5 in Pollard et al. (2016)).

The simulations are based on the Bedmap2 dataset (Fretwell et al., 2013), remapped to 16 km resolution, which corresponds to a total grounded modern Antarctic Ice Sheet volume of 56.85 m SLE (or $26.29 \times 10^6$ km$^3$). The ensemble mean at the end of the simulations (in the year 2000 or -0.05 kyr BP) underestimates the observed ice volume slightly by $0.6 \pm 3.5$ m SLE, or in terms of grounded ice volume by $0.7 \pm 1.7 \times 10^6$ km$^3$ (see Fig. 9). The histogram of score-weighted sea-level

anomalies at the end of all simulations can be adequately approximated by a normal distribution (Fig. 10 d).

### 3.2.1    Comparison of sea-level estimates to previous studies and model observations

For the maximum Antarctic ice volume at the Last Glacial Maximum, the inferred ensemble range of 5.3 - 13.5 m SLE (or 4.5 - $8.5 \times 10^6$ km$^3$) is at the upper range found in the recent literature (Fig. 11).

Model reconstructions are basically based on four different models: "Glimmer" (Rutt et al., 2009), "PSU" (or PennState3D) from Penn State University (Pollard and DeConto, 2012), "ANICE" from Utrecht University (De Boer et al., 2013) and, as in this study, the Parallel Ice Sheet Model (PISM; Winkelmann et al., 2011).

The modeled range between Last Glacial Maximum and present-day ice volume by Whitehouse

et al. (2012a) is about $5.0 \times 10^6$ km$^3$ (or 7.5 - 10.5 m ESL, eustatic sea-level based on volume above flotation), who ran a GIA-model based on a prescribed ice sheet history from different Glimmer simulations at 20 km resolution. Golledge et al. (2012) estimated a range of about $2.7 \times 10^6$ km$^3$ (or 6.7 m SLE using a conversion factor of 2.47) from PISM on a 5 km grid. For the same model and resolution Golledge et al. (2013) found about $3.35 \times 10^6$ km$^3$ (or 8.3 m SLE using a conversion factor

of 2.47). Briggs et al. (2014) estimated a range between 2.2 and $5.7 \times 10^6$ km$^3$ (or 5.6 - 14.3 mESL, using a conversion factor of 2.52) from PSU simulations for 40 km resolution. Maris et al. (2014) used ANICE on 20 km resolution and inferred around 3.8 - $4.8 \times 10^6$ km$^3$ (or 9.4 - 12.0 m s.l.e. using





a conversion factor of 2.5). For the same model and resolution Maris et al. (2015) found around
3.5 - 5.2 $\times 10^6$ km$^3$ (or 8.4 - 12.5 m s.l.e. using a conversion factor of 2.4). A much higher LGM
volume of about 5.8 $\times 10^6$ km$^3$ (or 10.5 m SLE ice volume above flotation relative to observations)
was retrieved from PISM on 15 km resolution by Golledge et al. (2014). Around 1.6 - 4.8 $\times 10^6$ km$^3$
(or 5 - 10 m ESL eustatic seal-level change for WAIS only, or 4 - 12 m ESL using a conversion factor
of 2.48) was found from the PSU model on 20 km resolution in Pollard et al. (2016) and around
2.8 - 4.1 $\times 10^6$ km$^3$ (or 3 - 8 m GMSL global mean sea-level change, or 6.9 - 10.2 m GMSL using a
conversion factor of 2.48) from PSU on a 20 km grid in Pollard et al. (2017). In the Large Ensemble
by (Pollard et al., 2017) ice volume change since LGM is somewhat biased to comparably low
values, as the used scoring algorithm pushed the ensemble to rather slippery basal sliding coefficient
on modern ocean beds (personal communication with Dave Pollard).

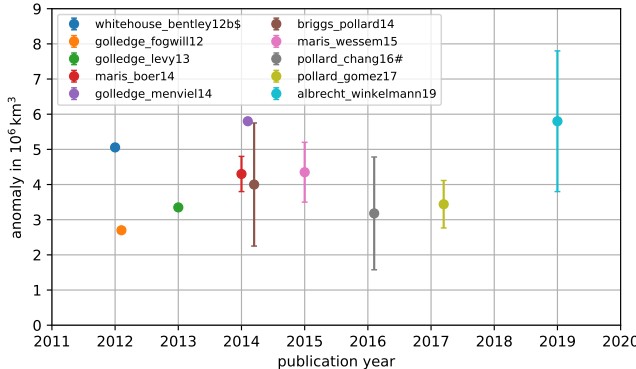

Figure 11: Ice volume anomaly between Last Glacial Maximum as compared to present in recent
modeling studies in units of $10^6$ km$^3$. Note that the study by Pollard et al. (2016) only considers
the West Antarctic subdomain in their analysis (grey). Golledge and colleagues and this study used
PISM, Maris and colleagues used ANICE, Whitehouse and colleagues used GLIMMER and Briggs
and Pollard and colleagues used PennState3D as model. Be aware, that ice volume estimates are
based on different ice densities in the different models.

### 3.3   Best-fit ensemble simulation

The best-fit ensemble member simulation (no. 6165, see Table 1) provides an Antarctic Ice Sheet
configuration for the present day, which is comparably close to observations. The present-day ice
volume of the West Antarctic Ice Sheet is in our ensemble simulations generally overestimated
(by around 25%), while the much larger East Antarctic Ice Sheet volume is rather underestimated
(by around 5%). Part of the overestimation can be explained by the relatively coarsely resolved
topography of the Antarctic Peninsula and weakly constrained basal friction in the Siple Coast area.





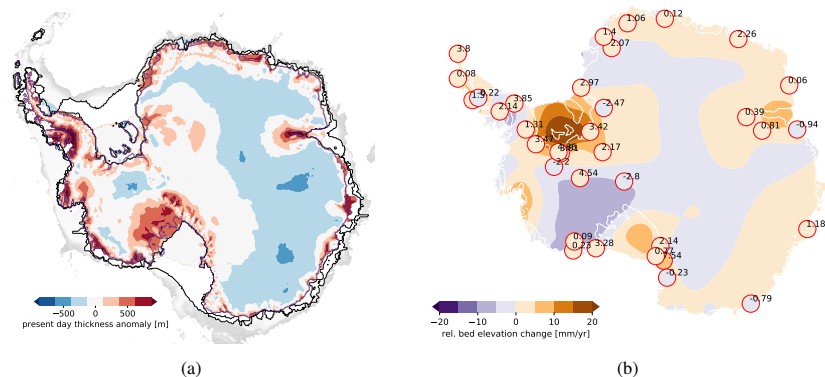

(a)            (b)

Figure 12: (a) Present-day ice thickness anomaly of best fit ensemble simulation with respect to observations (Fretwell et al., 2013), with the continental shelf in grey shades. Blue line indicates observed grounding line, while black lines indicate modeled grounding line and calving front. Large areas of the East Antarctic Ice Sheet are underestimated in ice thickness, while some marginal areas along the Antarctic Peninsula, Siple Coast and Amery Ice Shelf are thicker than observed, with a total RMSE of 266 m.

(b) Modeled uplift (violet) and depression (brown) at present-day state as compared to uplift rates from recent GPS measurements (Whitehouse et al., 2012b) in 35 locations (in units mm/yr).

This results in a RMSE of ice thickness of 266 m (see Fig. 12 a), a RMSE of grounding line distance of 67 km (see Fig. 13) and a RMSE for surface velocities of 66 m/yr (see Fig. 14). The best-fit simulation also reproduces the general pattern of observed modern isostatic adjustment rates of more than 10 mm yr$^{-1}$ (see Fig. 12 b) with highest uplift rates in the Weddell and Amundsen Sea Region
in agreement with GIA model reconstructions (cf. Argus et al., 2014, Fig. 6). In contrast to these GIA reconstructions, our best-fit simulation shows depression rather than uplift in the Siple Coast regions as grounded ice is still re-advancing and hence adding load.





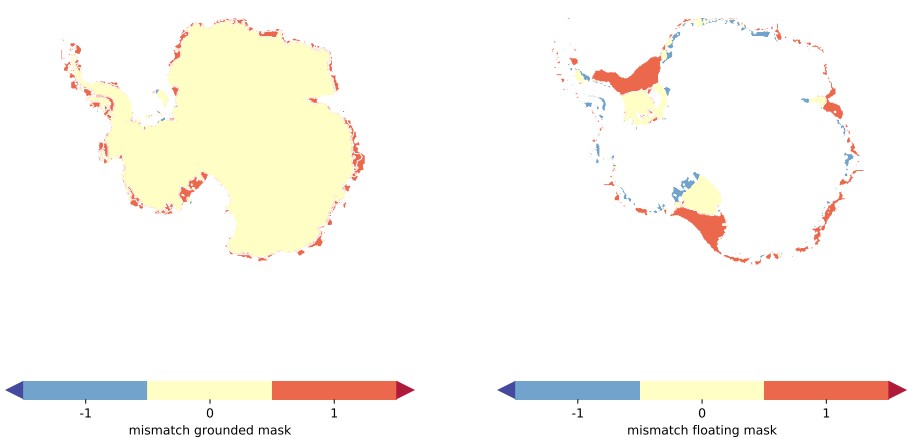

Figure 13: Comparison of present-day grounded (left) and floating (right) ice extent in best fit ensemble simulation with respect to observations (Fretwell et al., 2013). Yellow color indicate a match of simulation and observations, orange means grounded/floating in model but not in observations, and blue vise versa. Root-mean-square distance of modeled and observed grounding line is 67 km.

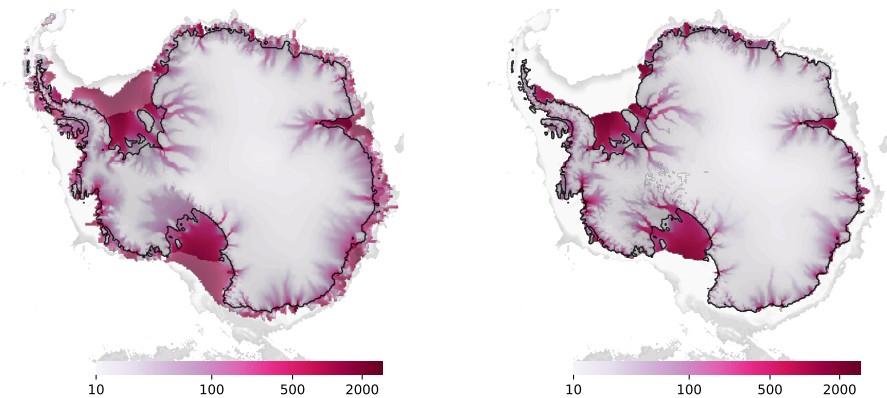

Figure 14: Comparison of present-day surface velocity in best fit ensemble simulation (left) with respect to observations (right, Rignot et al., 2011). Red shading indicates regions of fast ice flow with ice shelves and far-inland reaching ice streams. RMSE for surface velocities is 66 m/yr.

As the sea-level curve of the best-score simulation is close to the ensemble mean (Fig.9), a more detailed look into subsequent snapshots since the last glacial termination helps to identify periods

of comparably strong changes. Before 10 kyr BP our best-fit simulation shows extended ice sheet flow towards the outer Antarctic continental shelf edges, with more than 2,000 m thicker ice than today in the basins of the largest modern ice shelves (Ross, Weddell, Amery and Amundsen), while





the inner East Antarctic Ice was a few hundred meters thinner than today (see Fig. 15). At glacial
maximum around 15 kyr BP our simulations agree well with reconstructions by the RAISED Con-

sortium (Bentley et al., 2014, cf. Fig. 7 a). Last Glacial Termination, and hence major ice sheet
retreat initiates in the Ross and Amundsen sector in the best-score simulation around 9 kyr BP, in the
Amery sector around 8 kyr BP and in the Weddell Sea Sector at around 7 kyr BP. Recent proxy-data
reconstructions from the eastern Ross continental shelf suggested initial retreat not before 11.5 kyr
BP (Bart et al., 2018), likely around 9-8 kyr BP (Spector et al., 2017), which is consistent with our

model simulations. In the reconstructions by the RAISED Consortium most of the retreat in the Ross
Sea Sector (almost up to present-day grounding line location) occurred between 10 kyr BP and 5 kyr
BP, while major retreat in the Weddell Sea Sector happened before 10 kyr BP in scenario A and after
5 kyr BP in scenario B (Bentley et al., 2014, cf. Fig. 7 b,c). In our simulations a Holocene minimum
ice volume is reached in the late Holocene with slight re-advance and thickening in the Siple coast

and Bungenstock ice rise until present-day (see Fig. 16). This regrowth signal cannot be inferred
from RAISED reconstructions with only 5 kyr snapshots (Bentley et al., 2014). The corresponding
mass change agrees well with sea-level relevant volume change with about 0.07 mm/yr SLE ice sheet
regrowth (or 60 Gt/yr) in the last 3000 years, see Fig. 17.





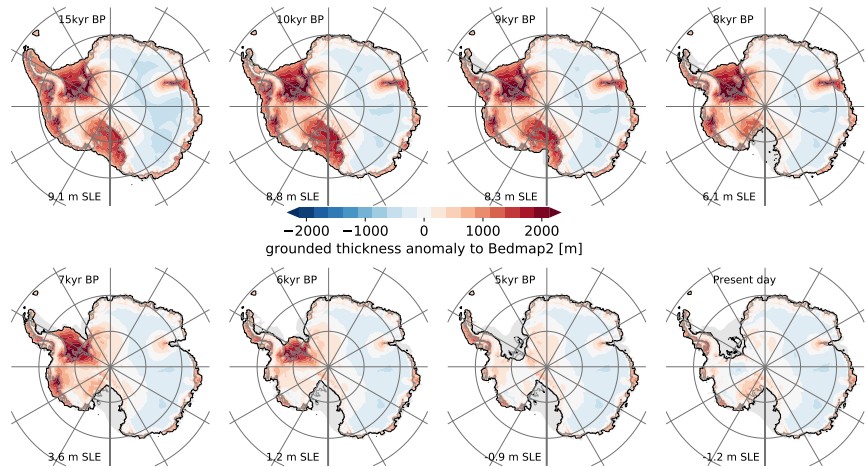

Figure 15: Snapshots of grounded ice thickness anomaly to present-day observations Fretwell et al. (2013) over last 15 kyr of best-fit simulation. At LGM state grounded ice extends towards the edge of the continental shelf with much thicker ice, mainly in West Antarctica. Retreat of the ice sheet occurs first in the Ross basin between 9 and 8 kyr BP, followed by the Amery basin around 1 kyr later and the Amundsen and Weddel Sea basin between 7 and 5 kyr BP. East Antarctic Ice Sheet thickness is underestimated throughout the deglaciation period (light blue). Compare Fig. 2 in Golledge et al. (2014).

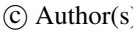


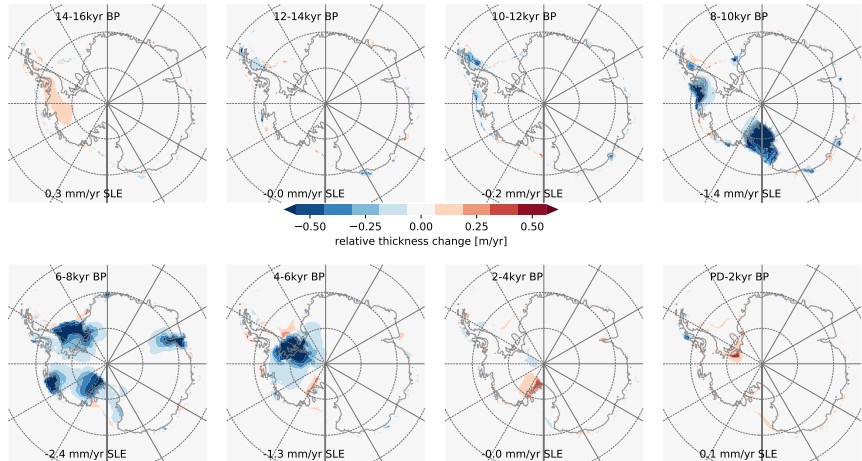

Figure 16: Snapshots of relative ice thickness change rates every 2 kyr over last 16 kyr of best-fit simulation. Deglaciation starts in the Ross and Amundsen Sector after 10 kyr BP with a mean change rate of -1.4 mm/yr SLE followed by the Amery and Weddell Sea Sector after 8 kyr BP with mean change rates of up to -2.4 mm/yr SLE (with peaks of up to -5 mm/yr SLE at 7.5 kyr BP). In the late Holocene period since 4 kyr BP the best fit simulation shows some thickening in the Siple Coast and in the Bungenstock Ice Rise corresponding to about +0.1 mm/yr SLE. Compare Fig. 4 in Golledge et al. (2014).

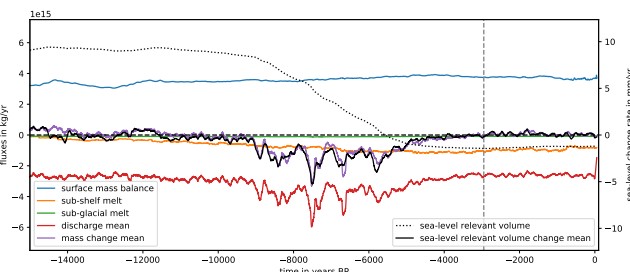

Figure 17: Mass fluxes over the last 15 kyr for the best-fit simulation (left axis), with the sum of surface (blue) and basal mass balance (orange and green) and discharge (100 yr running mean in red) as mass change (violet). Mass change agrees well with sea-level relevant volume change (100 yr running mean in black, right axis) with about 0.07 mm/yr SLE ice sheet regrowth (or 60 Gt/yr) in the last 3000 years (indicated by dashed vertical line).



### 3.4 Comparison to previous large ensemble study

Our study follows closely the Large Ensemble analysis method by Pollard et al. (2016) for simulation results of the West Antarctic Ice Sheet with the PSU model for 20 km grid resolution. We performed our analysis for the entire Antarctic Ice Sheet for 16 km grid resolution using PISM and four different model parameters (Sect. 2.1). Pollard et al. (2016) span their ensemble with parameters that involve mainly oceanic properties (ice shelf melting and calving) or properties of modern ocean-bed areas

(sliding over ice-shelf basins and bedrock relaxation time), while other parameters that affect the modern grounded ice areas are sufficiently constrained by earlier studies.

For the bedrock response we chose upper mantle viscosity as one of the parameters in our ensemble with maximum scores around values of VISC $= 5 \times 10^{21}$ Pa s. This corresponds to a rebound time scale of a few thousand years, which is in line with the findings in Pollard et al. (2016) using a

simplified Earth model (ELRA). Pollard et al. (2017) used the same analysis tools but additionally varied the vertical viscoelastic profiles of the Earth within a gravitationally self-consistent coupled Earth-sea level model. They found only little difference in simulated glacial to modern ice volumes for different viscosity profiles bounded between $1 \times 10^{19}$ Pa s and $5 \times 10^{21}$ Pa s.

Instead of a friction coefficient underneath the modern ice shelves we chose the sliding exponent

as uncertain parameter for the entire Antarctic Ice Sheet. Regarding sensitivity of the simulated ice volumes for variation of the minimum till friction angle underneath the modern ice shelves see our discussion in Albrecht et al. (2019) and the Appendix A.

In their ensemble analysis Pollard et al. (2016) included an iceberg calving parameter. Our 'eigencalving' model provides a fair representation of calving front dynamics independent of the climate

conditions (Levermann et al., 2012). Variations of the 'eigencalving' parameter show only little effect on sea-level relevant ice volume (see Albrecht et al. (2019)). Pollard et al. (2016) also included an uncertain parameter for ice shelf melt in their analysis. As we used the PICO model (Reese et al., 2018) that includes physics to adequately represent melting and refreezing also for colder-than-present climates, we have chosen other parameters in our ensemble, that are more relevant for

the ice volume history of the eastern part of the Antarctic Ice Sheet, namely ESIA and PREC (see Sect. 2.1).

## 4 Conclusions

We have run a Large Ensemble of simulations over the last two glacial cycles and have applied a simple averaging method with full factorial sampling similar to Pollard et al. (2016). Although the Large

Ensemble method is limited to a comparably small number of values for each parameter and hence the retrieved scores are somewhat blocky (as compared to advanced techniques that can interpolate in parameter space) we still recognize a general pattern and parameter combination clusters that provide best model fits to present-day observations and paleo records. For the four sampled parame-



ters best fits are found for comparably small mantle viscosity around VISC = 5–25 × $10^{20}$ Pa s, rather

linear relationships between sliding speed and till strength (with exponents PPQ = 0.75–1.0), no or
only small enhancement of the SIA derived flow speed (with ESIA = 1–2) and for rather high rates
of relative precipitation change with temperature forcing (PREC > 5 %/K). The five best-score en-
semble members fall within this range. In comparison to the best-fit member (VISC = 5 × $10^{20}$ Pa s,
PREC = 7 %/K, PPQ = 0.75, ESIA = 2) more sliding (PPQ = 1) or slower ice flow (ESIA = 1) can

compensate for relatively dry climate conditions in colder climates for high PREC values, which is
associated with smaller ice volumes and hence smaller driving stresses. Strongest effects of varying
ESIA and PREC parameters are found for the much larger East Antarctic Ice Sheet volume, while
PPQ and VISC have most pronounced effects in the West Antarctic Ice Sheet dynamics in terms
of grounding line migration and induced changes in ice loading. Grounding line extends at Last

Glacial maximum for nearly all simulations to the edge of the continental shelf. The onset and rate
of deglaciation, however is very sensitive to the choice of parameters and boundary conditions. Due
to the comparably coarse resolution and the high uncertainty that comes with the strong non-linearity
(sensitivity) of the system we here discuss rather general patterns of ice sheet histories than exact
numbers.

The score-weighted likely range (one standard deviation) of our reconstructed ice volume histo-
ries suggest that the Antarctic Ice Sheet has contributed 9.4 ± 4.1 m SLE (6.5 ± 2.0 × $10^6$ km³) to
the global mean sea level since the Last Glacial Maximum at around 15 kyr BP and reproduces the
observed present-day grounded ice volume with an score-weighted anomaly of 0.6 ± 3.5 m SLE
(0.7 ± 1.7 × $10^6$ km³). Our ensemble-mean lies at the upper range of most previous studies, except

for the large ensemble study by Pollard et al. (2017) with only 3–8 m SLE since LGM, as their
score algorithm favored the more rigid and hence thinner ice sheet configurations. Our reconstructed
ensemble range (1$\sigma$) is comparably large with up to 4.3 m SLE (or 2.0 × $10^6$ km³), which can be
explained with a high model sensitivity (Albrecht et al., 2019), a comparably large range of sam-
pled parameters and of course due to the choice of the aggregated score scheme (the unweighted

ensemble range would be up to 5.4 m SLE). A similar large range of is found in Briggs et al. (2014)
with 4.4 mESL (or 1.8 × $10^6$ km³) but for a different definition of volume change. The onset of
deglaciation and hence major grounding line retreat occurs in our model simulations after 12 kyr BP
and hence considerably after MWP1a (≈14.3 kyr BP). Previous studies with PISM Golledge et al.
(2014) suggest that the oceanic forcing at intermediate levels can be of opposite sign as compare to

the surface forcing, as during the two millennia of Antarctic Cold Reversal following the MWP1a.

The PISM model results in Kingslake et al. (2018) are based on this study but have been published
before with an older model version (see data and model code availability therein). Meanwhile we
have improved the bedrock model, which accounts for changes in the ocean water column induced by
variations in in bed topography or sea-level changes. Regarding the grounding line migration along

a trough through the Weddell Sea sector we found that Bungenstock Ice Rise is a key region and in





only about 20% of the score-weighted simulations this region regrounded. In this study we used the Bedmap2 topography remapped to 16 km resolution without local adjustments. Our paleo simulation ensemble analysis with PISM provides model and observation calibrated parameter constraints for projections of Antarctic sea-level contributions. With the best-fit simulation parameters we have
participated in the initMIP-Antarctica model intercomparison (Seroussi et al., 2019, PISMPAL3).

**Code and data availability**

The PISM code used in this study can be obtained from https://github.com/talbrecht/pism_pik/tree/ pism_pik_1.0 and will be published with DOI reference. Results and plotting scripts are available upon request and will be published in www.PANGAEA.de. PISM input data are preprocessed using
https://github.com/pism/pism-ais with original data citations.

**Appendix A:  Ensemble of basal parameters**

In the sensitivity analysis of parameters in a companion paper (Albrecht et al., 2019), we found that the basal sliding parameterization in conjunction with the sub-glacial hydrology scheme show very diverse simulated ice volume histories for a plausible range of unconfined parameter values. We have
chosen the parameter PPQ (Sect. 3.1) as only representative of basal processes uncertainties for the large ensemble analysis.

Here we want to span a sub-ensemble including three other relevant basal parameters. The four parameters and sampled values used in the sub-ensemble analysis are:

– PHIMIN: Minimal till friction angle on the continental shelf, mainly underneath modern ice shelves, where sandy sediments are prevalent (friction coefficient on the continental shelf has been chosen as one of the ensemble parameters in Pollard et al. (2016, 2017)). The tangens of till friction angle enters the Mohr-Coulomb-yield stress criterion. Sampled values are $0.5°$, $1°$, $2°$ and $3°$.

– TWDR: The decay rate of the effective water content within the till layer using the non-conserving hydrology model, while basal melt adds water up to a certain threshold. Sampled value are $0.5$ mm/yr ($1.55 \times 10^{-11}$ m/s), $1$ mm/yr ($3.1 \times 10^{-11}$ m/s), $5$ mm/yr ($15.5 \times 10^{-11}$ m/s) and $10$ mm/yr ($31 \times 10^{-11}$ m/s).

– FEOP: For this fraction of the effective overburden pressure (for details see Bueler and van
Pelt, 2015, Sect. 3.2), excess water will be drained into a transport system in the case of saturated till. Sampled values are 1%, 2%, 4%. 8% and 32%.

– PPQ: as in the large ensemble (see Sect. 3.1)





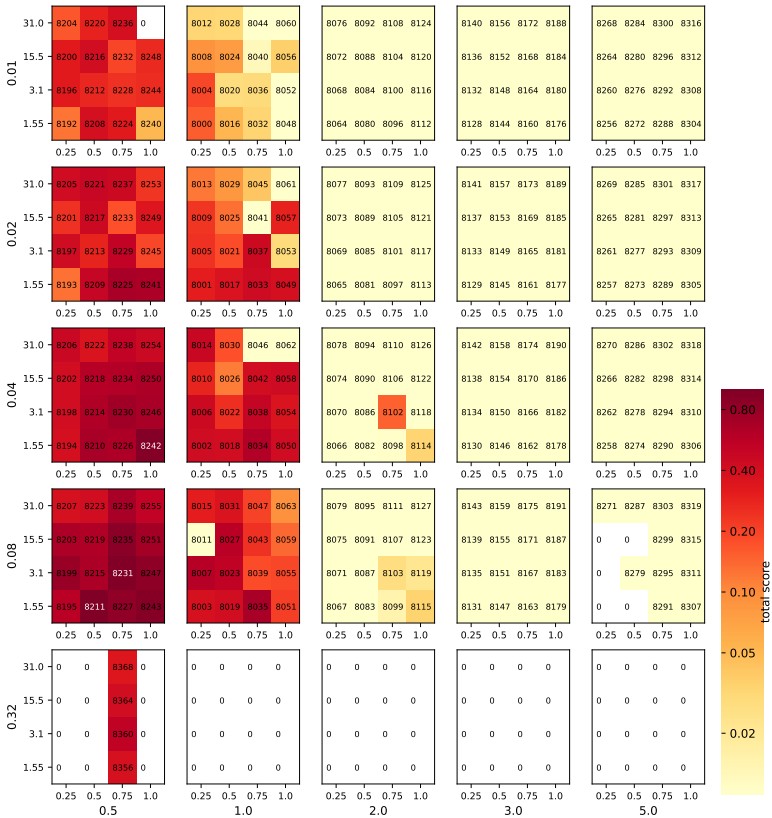

Figure A.1: Aggregated score for 318 ensemble members (4 model parameters, 4-5 values each) showing the distribution of the scores over the full range of plausible basal parameter values. The score values are computed versus geologic and modern data sets, normalized by the best score in the ensemble, and range from <0.01 (bright yellow, no skill) to 1 (dark red, best score) (cfs. Pollard et al., 2016, Figs. 2 + C1), on a logarithmic color scale. The four parameters are the effective overburden pressure fraction (outer y-axis), the minimal till friction angle on the continental shelf (outer x-axis), the tillwater decay rate (inner y-axis) and the power-law sliding pseudoplasticity exponent PPQ (inner x-axis). Only four ensemble scores are shown for 32% of effective overburden pressure fraction to ascertain that aggregated scores decline.

In the basal sub-ensemble we find even better scores than for the best fit parameter combination in the large ensemble (here no. 8102, see Fig. A.1), in particular for smaller minimal till friction

angles PHIMIN = 0.5–1°. Best scores are also found for rather high values of the fraction of the effective overburden pressure at which excess water drains, here FEOP = 4–16%. These values are




higher than those used in the large ensemble. However, best fit to the nine constraints are found for
the basal ensemble in the middle range of PPQ = 0.5–0.75 and the lower range of till water decay
rates of TWDR = 0.5-1 mm/yr ($1.55$–$3.1 \times 10^{-11}$ m/s), which agrees with the best fit values of large

ensemble. The LGM volume of the best fit simulation of the basal ensemble is similar to the best
fit simulation of the large ensemble (cf. Figs. A.3 and 15), however deglacial retreat occurs a few
thousand years earlier for lower PHIMIN.

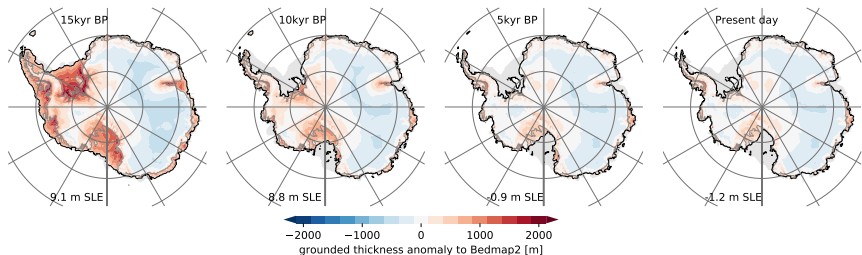

Figure A.2: Snapshots of grounded ice thickness anomaly to present-day observations (Bedmap2;
Fretwell et al., 2013) over the last 15 kyr for best-fit simulation in the basal ensemble. At LGM state
grounded ice extends towards the edge of the continental shelf, with much thicker ice than present
mainly in West Antarctica. Retreat of the ice sheet initiates between 12 and 11 kyr BP and halts
already latest 8 kyr in all large ice shelf basins of Ross, Weddell Sea, Amery and Amundsen Sea.
East Antarctic Ice Sheet thickness is underestimated throughout the deglaciation period (light blue).
Compare Fig. 2 in Golledge et al. (2014).





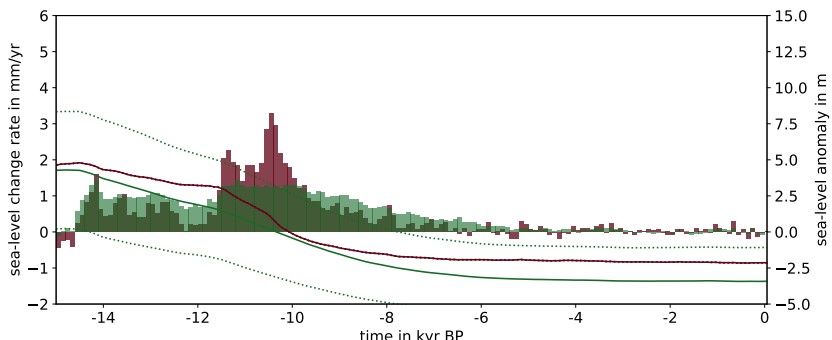

Figure A.3: During deglaciation the score-weighted ensemble mean (green) shows most of the sea-level change rates between 14.5 kyr BP (MWP1a) and 8 kyr BP with mean rates around 1 mm yr$^{-1}$, while the best-score simulation (red) reveals rates of sea-level rise of up to 4 mm yr$^{-1}$ (100 yr bins) in the same period (cf. Golledge et al., 2014, Fig. 3 d). In contrast to the Large Ensemble including climate and Earth model uncertainty (cf. Fig. 9c) the basal ensemble shows a much earlier retreat and no regrowth during the late Holocene.

*Acknowledgements.* Development of PISM is supported by NASA grant NNX17AG65G and NSF grants PLR-

1603799 and PLR-1644277. The authors gratefully acknowledge the European Regional Development Fund (ERDF), the German Federal Ministry of Education and Research and the Land Brandenburg for supporting this project by providing resources on the high performance computer system at the Potsdam Institute for Climate Impact Research. Computer resources for this project have been also provided by the Gauss Centre for Supercomputing/Leibniz Supercomputing Centre (www.lrz.de) under Project-ID pr94ga and pn69ru. T.A. is

supported by the Deutsche Forschungsgemeinschaft (DFG) in the framework of the priority program "Antarctic Research with comparative investigations in Arctic ice areas" by grant LE1448/6-1 and LE1448/7-1. We thank Dave Pollard for sharing ensemble analysis scripts and for valueable discussions.



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
