# Peer review of "Glacial cycles simulation of the Antarctic Ice Sheet with PISM - Part 2: Parameter ensemble analysis"

_The Cryosphere, 2019_

## Referee Comment (RC1) · Anonymous Referee #1 · 26 Jun 2019

**1   Overall assessment**

This study presents a large ensemble modelling of the Antarctic ice sheet over the last two glacial cycles with the PISM ice-sheet model. The ensemble reveals clusters of best fit parameters that are evaluated against a series of constraints related to the present-day ice sheet and glacio-geological evidence. Results of the best fit(s) reveal the deglaciation history of the Antarctic ice sheet (in line with previous results reported in Kingslake et al., 2018) and show the major ice loss after MWP-1A.

My first concern is the choice of the sensitivity parameters in the ensemble, which is

limited to four factors: ice softness (ESIA), sliding plasticity (PPQ), precipitation scaling (PREC) and mantle viscosity (VISC). Similar studies also explore the sensitivity of sub-shelf melting and how it relates to changes in far-field/continental shelf ocean and salinity in terms of oceanic forcing. Especially with respect to the explanation of MWP-1A, ocean forcing and its relation to sub-shelf melt may have played a crucial role (Golledge et al., 2014). Similar sensitivities have been explored in Pollard et al. (2016). Why are such parameters not taken into account, both sensitivity parameters within PICO, but also sensitivity to forcing, i.e. relation between atmospheric/ocean temperature forcing, for instance? As I understand from the paper, ocean temperature/salinity changes in the far field are not considered, neither through an offline ocean model, nor a parameterization that links atmospheric temperature change to oceanic temperature change. This is extremely important, as the conclusions with respect to the deglaciation do not take into account this sensitivity, hence show a large deglaciation pulse significantly later than the occurrence of MWP-1A. Many studies have shown the importance of the ocean in the dynamics of the Antarctic ice sheet, but neither the sensitivity (of PICO) or any ocean forcing has been investigated.

A related question is why choosing those four parameters (ESIA, PPQ, PREC and VISC) and not others? Have other studies or previous experience shown that these are the most sensitive/critical? Some explanation should be given.

A second concern is about the novelty of the study, that methodologically is heavily relying on Pollard et al. (2016) and is basically performing the same analysis. However, a clear rationale on the choice of the boundaries for the parameter changes is lacking. Moreover, as shown in Figure 4, clear clusterings in misfit show up and best fit results are generally found in a much smaller range of parameter values (basically the range of two values for each parameter. Therefore, it seems to me that a smaller range subsampling would lead to an improved fit, hence reduce the uncertainties of the whole ensemble.

**2 Specific remarks**

Line 153: in -> to

Line 180 and following: All scores are aggregated into one score, thereby giving them an equal weight. However, some constraints are more reliable than others. Would different weighing lead to different results? Is there a certain bias towards one or several parameters; in other words, what is the result if scores would be calculated separately? Does this lead to the same clustering? Which scores are more representative?

Line 256: intermediate values of mantle viscosity give the best results. However, these are values for the whole Antarctic continent and several studies show that there is a distinct contrast in mantle viscosity between WAIS and EAIS. Would this not explain the best score (mean of both extremes)?

Line 278-79: why high basal friction? The power of the friction law only determines how sliding scales with $\tau_b$.

Figure 4: see general remarks: clustering demonstrates that the sampling range is too large and can be refined.

Line 334: sub-surface melt: ambiguous, could point to melt occurring just below the surface. Using a term as sub-shelf melt is more appropriate.

Line 378: remove 'with' and add year of communication.

---

## Referee Comment (RC2) · Lev Tarasov (Referee) · 16 Jul 2019

Jul 15, 19 22:35

tt70.txt

Page 1/7

Glacial cycles simulation of the Antarctic Ice Sheet with PISM - Part 2: Parameter ensemble analysis Torsten Albrecht 1, Ricarda Winkelmann 1,2, and Anders Levermann 1,2,3

This part II submission of Albrecht et al examines a moderately sized ensemble of Antarctic glacial cycle runs with the PIK variant of the PISM ice sheet model. The ensemble runs are scored against a range of paleo and present-day (PD) constraints. The scored ensemble uses a reasonably state of the art model for ensemble glacial cycle contexts and adds to the litterature of what various models will do for past AIS glacial cycles. I therefore see the study potentially worthy of publication in TC given the current bar. At some point in the future I hope that model-based studies will have the requisite level of uncertainty quantification to enable much more meaningful inferences about past ice sheet evolution. However even with the current bar, a number of significant deficiencies need to be addressed.

The experimental design has some significant problems that are not even discussed. The study only using 4 ensemble parameters. Briggs et al, (TC, 2013) for instance, have 12 ensemble parameters just for the climate forcing and 31 ensemble parameter in total. At least 1 of the 5 temperature related ensemble parameters in that study (Tmix1) was one of the most sensitive ensemble parameters (with generally more sensitivity to this than to the precipitation related parameter (PdeselevEXP) that best corresponds to the sole climate forcing parameter (PREC) in this submission. The lack of an ensemble parameter relevant to the temperature forcing is especially problematic given the stated context of providing a distribution of present-day ice sheet states for initializing future projection runs. That state will depend significantly in the 3D temperature field of the ice sheet, the uncertainty of which is not probed in this study. Ideally this would be remedied, but that would be a major endeavor. At the very least I expect a clear and complete discussion of model and experimental design weaknesses and associated relevance to given results. A summary of this should also be in the conclusions.

Another major omission is a comparison of the ensemble results against the paleo data constraints. The chosen normalization of all score components against median scores means that the scores do not convey any information about absolute model fit to paleo data, only relative fit. It is therefore incumbent that the complete set of ensemble fits to paleo constraints be explicitly shown and discussed, eg as done in Briggs et al, 2014.

Furthermore, there are a number of claims and statements (detailed below) made that I find are indefensible, misrepresentative, and/or incorrect.

**Specific comments**

Large Ensemble of 256
-> Ensemble of 256
**with eg Briggs et al using a 2000 member ensemble, you can hardly call 256 a**
**"Large Ensemble".**

"The model is calibrated against..."

Monday July 15, 2019

tt70.txt Jul 15, 19 22:35 Page 2/7 -> The model is scored against... # scoring a moderately sized ensemble is not calibration # what is the SSA enhancement factor value? It is never explicitly given # in this study nor in the PART I of this submission. This analysis further constrains relevant model and boundary parameters by revea ling clusters of best fit parameter combinations. # Isn't that already previously stated in different words: "The model is calibrated[scored] against..." Our Large Ensemble analysis also provides well-defined parametric uncertainty bounds and a probabilistic range of present-day states th at can be used for 20 PISM projections of future sea-level contributions from the Antarctic Ice She et. # Kind of meaningless. I can think up a dozen metrics that would provide "welldefined # parametric uncertainty bounds", each with different resultant ranges. nonconserving sub-glacial hydrology model # as from review of part I: pretty crude to call this a model -> # Here we use the non-conserving sub-glacial hydrology parametrization # Missing brief (eg 1 sentence) description of bed thermal model. Sub-shelf melting in PISM is calculated via PICO (Reese et al., 2018) from salinity and temperature in the lower ocean layers on the continental shelf (Schmidtko et al., 2014) in 18 separate basins based on (Zwally et al., 2015) adjacent to the ice shelves around the Antarctic continent # the companion paper states that salinity was not varied: # "While salinity change over time in the deeper layers is neglected in this stu dy" # and this should be made clear here. use the Large Ensemble approach # Why is this capitalized? "the" makes no sense as there # are lots of large ensemble approaches. Furthermore, has already # stated above, this is not a large ensemble. This method yields as reasonable results for an adequately resolved parameter space as more advanced statistical techniques with means of interpolating results between sparsely separated points in multi-dimensional parameter space. # I would strongly dispute this since full-factorial sampling # restricts one to a relatively small number of ensemble parameters. # The cited Pollard et al (2016) paper used an ensemble of # WAIS only simulations for the last 30 kyr. Ie all ensemble # members had identical initial conditions and identical # time evolving ice boundary conditions at the junction with # the East Antarctic ice sheet. This is a far cry from applying # 4 ensemble parameters to the whole AIS for 2 glacial cycles.

Printed by Lev Tarasov

tt70.txt Page 3/7 Jul 15, 19 22:35 It covers uncertainties within the Earth model for values of 1e19, 5e19, 25e19 and 100e19 Pa s. # this study would benefit from better attention to relevant # litterature. While there is local support for viscosity as low as # 1e19 Pa s on the Antarctic Peninsula, there is no support for even # 5e19 over say the whole WAIS. Furthermore, the upper bound test # viscosity (and please use the more standard  $X10^{-21}$  units as # preferred in the GIA community) is half of the 95% "confidence" # upper bound of 2.0 X10^21 Pa s of Whitehouse et al, 2012 (GJI). This compilation also in150 cludes records of regional sea-level change (RSL), which has not been considered in this study since most of the sea-level signal is a result of the sea-level forcing with up to 140 m rather than the model's ice dynamical response expressed in terms of sea-level equivalents, as P ISM lacks a selfconsistent sea-level model # Since the RSL data for Antarctica is above present-day elevation, # the above statement as written is incorrect. The RSL data is the # signal, and dominance of a far-field sea-level forcing would result # in sealevel below present-day. Mean-square-error misfit to observed grounding line location for the modeled Ant arc135 tic grounded mask (ice rises excluded) using a signed distancel field # I don't understand what you mean by signed distance as RMSE would # only care about unsigned distance. Or do you mean what we do in my # group: also track mean (ie not RMSE) error and use that to assign a # signed value to the RMSE? 5. UPL: Mean-square-error model misfit to modern GPS-based uplift rates on rock outcrops at 35 individual sites using the compilation by Whitehouse et al. (2012b, Table 2) including individual observational uncertainty # Would be good to update the GPS data-set. Current GPS data versus # that approaching a decade old would make a significant difference in # observations and observational uncertainties. Then the individual score Si, j is normalized according to the median to # Why the mean versus the median? As in Pollard et al. (2016) we also assume that each data type is of equal importance to the overall score, avoiding the inter-data-type weighting used by Briggs and Tarasov (2013); Briggs et al. (2014), which would favor data types of higher spatio-temporal density # Would you still do this if say you only had 3 ELEV or EXT # datapoints? If all data were statistically independent, then one # would demand that data types of higher spatio-temporal density would # get more weight since in this case each and every datum should have # equal weight. You need to provide a better justification for this
**choice then just blind citation of previous studies.**

Monday July 15, 2019

Jul 15, 19 22:35

tt70.txt

The parameter ESIA enhances the shear-dominated ice flow and hence ice thickness # enhanced ice flow will not enhance (ie thicken) ice thickness but thin it For the upper range of mantle viscosities up to VISC = 1022 Pa s we find a normalized ensemble mean of 27% and 20%, # This contradicts what you previously indicated was an upper bound value of 100 X10^19 Pas on page 4. ??? This value is also used in the GIA model ICE-6G (Peltier et al., 2015) # kind of irrelevant since Peltier doesn't do regional tuning of Viscosity profi les. The best score ensemble members are found for intermediate mantle viscosities of VISC=5X10^20 Pa s and VISC=25X10^20 Pa s. # THis again contradicts the values given on page 4. Furthermore, # 25X10^20 Pa s is a high viscosity for the upper mantle (upper mantle is what # this half-space model best corresponds to) 3.2 Reconstructed sea-level histories # -> ice volume histories or sea-level contribution histories the ensemble mean ice volume is 1.0m SLE below modern with a score-weighted standard deviation of around 2.7m SLE (volume of grounded ice above 300 flotation in terms of global mean sea level equivalent as defined in Albrecht et al. (2019). # Should compare this to published (paleo data-based) inferences of the Eemian h igh-stand # as it makes it hard to fit current proxy-derived estimates for the Eemian high -stand given # constraints on what Greenland could have contributed. The LGM ice volume increases for lower PPQ, lower PREC and lower ESIA 315 values, while it seems to be rather insensitive to the choice of VISC # All these relations would be expected as such. As MWP1a initiated the Antarctic Cold Reversal (ACR) with about two millenia of colder su rface temperatures, 330 a freshening of surface waters leading to a weakening of Southern Ocean over turning, resulting in reduced Antarctic BottomWater formation, enhanced stratification and sea-ice exp ansion. # This is not a sentence. The modeled range between Last Glacial Maximum and present-day ice volume by Whi tehouse 360 et al. (2012a) is about 5.0X10^6 km3 (or 7.5 - 10.5m ESL, eustatic sea-level based on volume above flotation),... # There is no point in listing all the exact ranges here and then showing those ranges in fig 11

| Jul 15, 19 22:35                                                                                                                                                                                                                                        | tt70.txt                                                                                                                                                                                                                                                                                                                                                                                                              | Page 5/7                   |
|---------------------------------------------------------------------------------------------------------------------------------------------------------------------------------------------------------------------------------------------------------|-----------------------------------------------------------------------------------------------------------------------------------------------------------------------------------------------------------------------------------------------------------------------------------------------------------------------------------------------------------------------------------------------------------------------|----------------------------|
| <pre># Add the conversion eneral comparison.</pre>                                                                                                                                                                                                      | factors to the figure key and have the paragraph                                                                                                                                                                                                                                                                                                                                                                      | focus on g                 |
| Briggs et al. (2014)
**To be accurate PSU**
**response with radia**
**subshelf melt, basa**
**treatments. So only                                                                                                                                 | from PSU simulations for 40 km resolution**
+ full visco-elastic isostatic adjustment bedrocally layered earth viscosity profile + different
al drag, climate forcing, and calving
y PSU ice dynamics and thermodynamics.                                                                                                                                                                                      | k                          |
| Although the Large
445 Ensemble method is
arameter
**no it is not. A ful**
**large ensemble appr**
**Ensemble" is capita**
**ensemble as a "Larg**
**readers supposed to**
**modelling studies t**
**sampling scheme? Wh**
**ensembles? | In the studies that have O(10) or more larger                                                                                                                                                                                                                                                                                                                                                                         | for each p**
r
re      |
| <pre># fig 11 plot and cap # studies have non sy # states : "likely # between 5.6 and 14. # confidence >10 mESI # whitehouseBently12k # never discusses ice # whitehouseBently12a # and provides an unc # datapoint with no use</pre>                | otion: there needs to be a note that some of the
'mmetry distributions. Eg, Briggs et al, 2014
3 m equivalent sea level (mESL), and with less
" = 4.0 X 10^6 km^3 of ice. The
o datapoint is also problematic as that GJI paper
e volume or total sea level changes. The
a does (and is cited in the preceeding discussion
certainty range but the plot only shows a single
incertainty range | indicated                  |
| <pre># fig 12 please incre # use with ageing eye</pre>                                                                                                                                                                                                  | ease the font size of the colour key for those of es                                                                                                                                                                                                                                                                                                                                                                  |                            |
| <pre># fig 14: please use # comparison more dis # of a difference plo</pre>                                                                                                                                                                             | a higher contrast colour scale, to make the
scernable. Even better would be the addition
of to make clear what the differences are.                                                                                                                                                                                                                                                                             |                            |
| <pre># fig 15-17 are hard1 # of details. Eg, fig # refered to in the t</pre>                                                                                                                                                                            | y mentioned in the main text, with no considerat
gure 17 has 7 timeseries, not one of which is ind
text. So why is this figure in the paper?                                                                                                                                                                                                                                                                    | ion
ividually           |
| 3.4 Comparison to pre
**-> Comparison to Po**
**Your current title**
**and the subsequent**
**Briggs et al was a                                                                                                                                  | evious large ensemble study**
ollard et al. (2016) ensemble study
implies there was only one large ensemble
text implies it Pollard et al. (2016).
much larger ensemble study                                                                                                                                                                                                                               |                            |
| while other parameter
modern grounded ice a
**This is not a fact,                                                                                                                                                                                 | es that affect the areas are sufficiently constrained by earlier stu-**
at best state they claim this.                                                                                                                                                                                                                                                                                                               | dies                       |
| In their ensemble and
ameter. Our 'eigencal
model provides a fair                                                                                                                                                                                 | alysis Pollard et al. (2016) included an iceberg
.ving'
c representation of calving front dynamics indepe                                                                                                                                                                                                                                                                                                       | calving par
ndent of th |

Printed by Lev Tarasov

tt70.txt Page 6/7 Jul 15, 19 22:35 e climate 435 conditions (Levermann et al., 2012). # What does "fair" mean? Be precise. As we used the PICO model (Reese et al., 2018) that includes physics to adequately represent melting and refreezi ng also for colder thanpresent climates, we have chosen other parameters in our ensemble, # What about the large uncertainty is subshelf ocean temperature? Above you invo ke this # to explain the lack of any MWP1a signal compared to eg Golledge et al, 2014 comparably small mantle viscosity around VISC = 5-25X10^20 Pa s, # small compared to what? I wouldn't call these small upper Mantle viscosities f or Antarctica Due to the comparably coarse resolution and the high uncertainty that comes with the strong non-linearity (sensitivity) of the system we here discuss rather general patterns of ice sheet histories than exact numbers. # This non-linearity is another reason to increase the number of ensemble parame ters. Our ensemble-mean lies at the upper range of most previous studies, except 470 for the large ensemble study by Pollard et al. (2017) with only 3-8m SLE sin ce LGM, as their score algorithm favored the more rigid and hence thinner ice sheet configuration s. # incorrect in that half of your stated range is below the favoured # range of Briggs et al (2014) who state "The LGM ice volume excess relative to present-day is likely between 5.6 and 14. 3 m equivalent sea level (mESL), and with less confidence >10 mESL # Furthermore, your sentence contradicts itself as currently written. Previous studies with PISM Golledge et al. (2014) suggest # english and punctuation... In this study we used the Bedmap2 topography remapped to 16 km resolution without local adjustments # Does this actually belong in the "Conclusions"? provides model and observation calibrated parameter constraints for projections of Antarctic sea-level contributions # awkward and somewhat indecipherable. Do you just mean # "data-contrained projections of .. using PISM"? With the best-fit simulation parameters we have 490 participated in the initMIP-Antarctica model intercomparison (Seroussi et al ., 2019, PISMPAL3). # This is not a conclusion # appendix A : is referred two a couple of times, but without any # statement of what the takeaway from the appendix is.

Jul 15, 19 22:35

**tt70.txt**

Page 7/7

In the basal sub-ensemble we find even better scores than for the best fit param eter combination in the large ensemble (here no. 8102, see Fig. A. However, best fit to the nine constraints are found for the basal ensemble.

which agrees with the best fit values of large ensemble.

**The above two statements contradict each other**

---

## Author Response (AR1)

**Response to reviewers on "Glacial cycles simulation of the Antarctic Ice Sheet with PISM – Part 2: Parameter ensemble analysis" by Torsten Albrecht et al.**

We would like to thank the anonymous reviewer and Lev Tarasov for the very constructive criticism regarding our manuscript. These reviews have considerably improved the manuscript for which we are grateful. We were able to address all requests.

In order to facilitate the reading of this document, the referee's comments are given in blue and in **black** the author's response.

**Response to Anonymous Referee #1**
()

1 Overall assessment

This study presents a large ensemble modelling of the Antarctic ice sheet over the last two glacial cycles with the PISM ice-sheet model. The ensemble reveals clusters of best fit parameters that are evaluated against a series of constraints related to the present-day ice sheet and glacio-geological evidence. Results of the best fit(s) reveal the deglaciation history of the Antarctic ice sheet (in line with previous results reported in Kingslake et al., 2018) and show the major ice loss after MWP-1A.

My first concern is the choice of the sensitivity parameters in the ensemble, which is limited to four factors: ice softness (ESIA), sliding plasticity (PPQ), precipitation scaling (PREC) and mantle viscosity (VISC). Similar studies also explore the sensitivity of sub-shelf melting and how it relates to changes in far-field/continental shelf ocean and salinity in terms of oceanic forcing. Especially with respect to the explanation of MWP-1A, ocean forcing and its relation to sub-shelf melt may have played a crucial role (Golledge et al., 2014). Similar sensitivities have been explored in Pollard et al. (2016). Why are such parameters not taken into account, both sensitivity parameters within PICO, but also sensitivity to forcing, i.e. relation between atmospheric/ocean temperature forcing, for instance? As I understand from the paper, ocean temperature/salinity changes in the far field are not considered, neither through an offline ocean model, nor a parameterization that links atmospheric temperature change to oceanic temperature change. This is extremely important, as the conclusions with respect to the deglaciation do not take into account this sensitivity, hence show a large deglaciation pulse significantly later than the occurrence of MWP-1A. Many studies have shown the importance of the ocean in the dynamics of the Antarctic ice sheet, but neither the sensitivity (of PICO) or any ocean forcing has been investigated.

We thank the reviewer for pointing out important aspects of the parameter choice and implications for the last deglaciation. Some of these questions are actually touched in the first part of the study (Albrecht et al., 2019a), which certainly need to be better referenced in this second part. Find our detailed comments below.

A related question is why choosing those four parameters (ESIA, PPQ, PREC and VISC) and not others? Have other studies or previous experience shown that these are the most sensitive/critical? Some explanation should be given.

The choice of the four ensemble parameter is motivated in the companion study (Albrecht et al., 2019a), in which different parameter choices and boundary conditions are compared to a reference model ice volume history to gain some „prior model experience" and to determine most relevant parameters (for this metric) in each of the different model components (climatic forcing, basal sliding, ice creep and bedrock response). We agree, that this parameter choice is somewhat biased to the modeled total ice volume at LGM and present-day state, while other parameters may be more relevant for the onset and pace of deglaciation. We have added a paragraph on deficiencies of the study. Yet, enhancement factors, sliding coefficient and viscous relaxation time of the bedrock have been also typically varied in previous model studies (e.g., Pollard and DeConto, 2012; Maris et al., 2014; Quiquet et al., 2018). As PISM uses a more generalized basal sliding and bedrock deformation scheme, we have selected different uncertain parameters.

Regarding the reviewer's concern on the sub-shelf melt sensitivity and the MWP-1A, we can state that Pollard et al. (2016) was focussing mainly on ice-oceanic deglacial processes in the WAIS with other relevant parameters fixed, while we consider a broader range of sea-level relevant processes over a longer time scale, such as ice-internal and ice-atmospheric effects, covering both parts of the Antarctic Ice Sheet. Golledge et al. (2014) used an apparently more realistic ocean forcing (from an Earth System Model), but they state „that there is considerable uncertainty in the relationship between ocean temperature and ice-shelf melt". In fact, much of the oceanic uncertainty of previous models is considerably reduced in our PISM simulations as it uses the PICO module (Reese et al., 2018), in which two uncertain parameters have been constrained by observed melt rates. Of course, we do consider ocean temperature changes, in our case coupled to surface temperature forcing (see Sect. 4.3 in Albrecht et al., 2019a). However, this relationship cannot account for events such as the Antarctic Cold Reversal after MWP-1A, when surface and intermediate water temperatures became rather decoupled. Yet, we have tested our PISM-PICO model for an earlier warming signal in the deeper ocean layers (while the surface was warming at the same time) in the companion paper (see Sect. 5.2 in Albrecht et al. 2019a), which can cause earlier retreat, while we still do not find main deglaciation before MWP-1A. For these reasons we have selected PREC as climate uncertainty instead of an ocean-melt (or calving) related parameter, as it can potentially counteract the other more constrained climatic forcings (see Sect. 4.5 in Albrecht et al. 2019a), and thsi aspect may have been underestimated by previous model studies. We have added some more discussion on the limited parameter choice and consequences for the results in the revised manuscript.

A second concern is about the novelty of the study, that methodologically is heavily relying on Pollard et al. (2016) and is basically performing the same analysis. However, a clear rationale on the choice of the boundaries for the parameter changes is lacking. Moreover, as shown in Figure 4, clear clusterings in misfit show up and best fit results are generally found in a much smaller range of parameter values (basically the range of two values for each parameter. Therefore, it seems to me that a smaller range sub-sampling would lead to an improved fit, hence reduce the uncertainties of the whole ensemble.

Yes, we have been using very similar analysis (and visualization) tools as discussed in Pollard et al. (2016) to allow for better comparison. However, Pollard et al. (2016) used different parameterizations in their model and focussed mainly on ice-oceanic processes in the WAIS over the last 20kyr. We have hence chosen different parameters and parameter boundaries as motivated in the (first part) companion paper (Albrecht et al., 2019a). A refined analysis could likely provide better constrained (best fit) parameter ranges. But

the high uncertainty in the sea-level history is in fact a result of multiple best-score parameter ensemble members which show quite different sea-level histories.

Also, the best-fit parameters of our paleo study might be shifted slightly for higher spatial resolution, e.g. when performing short-term projections. In this analysis we wanted to consider the broader range of parameter values, covering the wide parameter range used also in other models (implying a rather coarse sampling) to gain a better understanding of (combined) parameter effects in the highly nonlinear model. This also serves as rough constraint for further ensemble simulations and projections with PISM.

**2 Specific remarks**

Line 153: in -> to

Thanks.

Line 180 and following: All scores are aggregated into one score, thereby giving them an equal weight. However, some constraints are more reliable than others. Would different weighing lead to different results? Is there a certain bias towards one or several parameters; in other words, what is the result if scores would be calculated separately? Does this lead to the same clustering? Which scores are more representative?

This is definitely true, the score aggregation hides lots of information on the individual data types. However, we did not use inter-data-type weighting, e.g. based on spatial and temporal volumes of influence of each data type, as done in previous studies (Briggs and Tarasov, 2013; Briggs et al., 2014). Here, we followed the arguments in Pollard et al. (2016), assuming that „each data type is of equal importance to the overall score, and that if any one individual score is very bad ($S_i ≈ 0$), the overall score S should also be $≈ 0$... if any single data type is completely mismatched, the run should be rejected as unrealistic, regardless of the fit to the other data types... The fits to past data, even if more uncertain and sparser than modern, seem equally important to the goal of obtaining the best calibration for future applications with very large departures from modern conditions". We will refer more clearly to these argumentation in the revised manuscript.
We have also added Supplementary Material with plots of individual paleo data misfits analogous to Briggs et al. (2014). If using the inder-data-type weighting and defining the score as weighted sum as in Briggs et al. (2014), the resultant distribution of best scores (here the smallest values) actually turns out to be very similar as for the product of individual scores, as shown in Fig. R1.

[Figure]

[Figure]

*Fig. R1: Aggregated scores as product of individual scores as in Pollard et al. (2016) and used in our study (best fit equals 1, log color scale), compared to the aggregated score as a result of a inter-data-type weighted sum, as in Briggs et al. (2014), with best fits for lowest scores.*

In fact, we can learn more about the model's response when discussing statistics on individual scores. Some of these information can be estimated from Fig. 2 or Fig. 5 and are discussed rather qualitatively in corresponding sections. We added ensemble standard deviation (Table R1) for each data type and some more discussion on the stasticial aspects to the revised manuscript. Find also Fig. R2 and Fig. R3 for comparison (analogous to the ones in the manuscript, but separated for individual data-type scores).

| | TOT | TOTE | TOTI | TOTDH | TOTVEL | TOTGL | TOTUPL | TROUGH | ELEV | EXT |
|---|---|---|---|---|---|---|---|---|---|---|
| MAD | 0.002 | 0.123 | 0.023 | 0.183 | 0.144 | 0.099 | **0.292** | 0.156 | 0.049 | 0.047 |
| SD | 0.082 | 0.156 | 0.035 | 0.190 | 0.179 | 0.126 | **0.300** | 0.204 | 0.075 | 0.072 |

*Table R1: Medan absolute deviation (MAD) and standard deviation (SD) for each data-type score, SD values are used in the revised manuscript.*

In the manuscript (Sect. 3.1) we have discussed for each ensemble parameter how best scores are related to individual data types, as shown in Fig. R2. We want to avoid additional figures in the manuscript, but we added a general comment to the Supplementary Material B:

*„The corresponding variability of each of the resultant normalized scores hence contribute different skills to the aggregated score (see Table 2). Generally, grounding-line related (TOTE, TOTGL, THROUGH) and ice volume-related data-types (TOTDH) show similar individual score patterns (not shown here) with ensemble standard deviations of 0.1-0.2. In the aggregated score this patterns becomes even more pronounced, while paleo scores (ELEV and EXT) and ice shelf extent (TOTI) show only little variation (<0.1) among the ensemble, and hence only little effect in the aggregate score pattern."*

[Figure]

*Fig. R2: Individual data-type scores for all 256 ensemble members, as in Fig. 1 in the manuscript, but with linear color scale. Scores in individual data-types are normalized by median.*

[Figure]

Fig. R3: Individual data-type mean scores for six possible pairs of parameter values, as in Fig. 4 in the manuscript.

The score pattern is also shaped by the uplift-related individual score (TOTUPL), that shows the highest ensemble standard deviation of 0.3 (Table R1) with a clear tendency towards higher VISC values, (see Fig. R3) probably a result of lower sensitivity to fluctuations in grounding line location. In contrast, the velocity-related individiual score (TOTVEL) with ensemble standard deviations of 0.2, favors lower VISC values, probably a result of more advanced grounding line location, which implies lower ice shelf velocities and hence lower chance for misfit (see Fig. R3). In the product formulation of the aggregated score such a reverse pattern can lead to highest total values for intermediate parameter values for VISC. We have discussed this aspect in the revised manuscript:

*„As mantle viscosity determines the rate of response of the bed to changes in ice thickness a low viscosity corresponds to a rather quick uplift after grounding line retreat and hence to a retarded retreat, which corresponds to a rather extended present-day state. This implies smaller ice shelves with slower flow and less velocity misfit, such that also TOTVEL favors small VISC values. In contrast, a trend to rather high mantle viscosities in the aggregated score stems mainly from the misfit of present-day uplift rates expressed as data-type score TOTUPL, probably due to reduced sensitivity to fluctuations in grounding line location. High mantle viscosities involve a slow bed uplift and grounding line retreat can occur faster. More specifically, in the partially over-deepened ice shelf basins, which have been additionally depressed at the Last Glacial Maximum by a couple of hundred meters as compared to present, grounding line retreat can amplify itself in terms of a regional Marine Ice Sheet Instability (Mercer, 1978; Schoof, 2007; Bart et al., 2016). In fact, the best score ensemble members are found for intermediate mantle*

*viscosities of VISC =$0.5×10^{21}$ Pa s, and VISC =$2.5×10^{21}$ Pa s. This could be a result of the product formulation of the aggregated score, in which individual data types scores favor opposing extreme values."*

[Figure]

*Fig. R4: Map of misfit of modeled modern surface velocity (related to TOTVEL) in four ensemble members with different VISC values indicated in labels (but otherwise identical parameters).*

Line 256: intermediate values of mantle viscosity give the best results. However, these are values for the whole Antarctic continent and several studies show that there is a distinct contrast in mantle viscosity between WAIS and EAIS. Would this not explain the best score (mean of both extremes)?

This is a good question. Recent literature suggest comparably small values for the oceanic WAIS plate. As most of the ice volume and grounding line changes occur in WAIS, one would suggest that this regions also leaves the strongest imprint on the individual-scores, that are related to the VISC parameter.

As already mentioned above, in our ensemble it is the TOTVEL data type which favors lower values, likely related to the grounding line location and ice shelf extent (see also Fig. R4), while TOTUPL actually favors large VISC values, which might actually be related to better scores for the EAIS part, where lower bedrock sensitivity and lower measurement uncertainty leads to lowest misfits (see Fig. R5).

[Figure]

*Fig. R5: Misfit of modeled present-day bedrock change rates to GPS measurements (related to TOTUPL) around the Antarctic continent for four different VISC values. Insets show location of PGS sites and map of bedrock change.*

Line 278-79: why high basal friction? The power of the friction law only determines how sliding scales with тb .

Thanks for pointing out this imprecise formulation. Basal shear stress $\tau_b$ balances the driving stress within the SSA stress balance. As in the PISM friction law $u_0$ is considered as reference velocity (Eq. 2), such that for $q>0$ slower flowing upstream regions experience reduced basal shear stress, while fast flowing regions downstream are subjected to increased basal shear stress. Thus, reducing $q$ from 0.75 to 0.25 produces slower flow in the interior and faster ice stream flow. We omitted this confusing aspect in the paragraph in the revised manuscript:

*„In about 10% of the score-weighted simulations grounding line remains at the extended position without significant retreat, linked to  an efficient negative feedback on grounding line motion related to a fast responding bed (low VISC)."*

Figure 4: see general remarks: clustering demonstrates that the sampling range is too large and can be refined.

As already stated above we intended to cover a broad range of parameter values typically (and plausibly) used in other ice sheet modeling studies for better comparison (ESIA, PPQ, VISC) and to gain a better understanding of the actual (combined) effects of

parameters on the ice sheet dynamics. For follow-up projections (with higher resolution) a similar score scheme may be used, but for different (more recent) data-types in terms of hindcasting, with more refined parameter ranges.

Line 334: sub-surface melt: ambiguous, could point to melt occurring just below the surface. Using a term as sub-shelf melt is more appropriate.

Thanks, has been changed accordingly in the revised manuscript.

Line 378: remove 'with' and add year of communication.

Changed to „(personal communication Dave Pollard, 2017)".

**Response to Referee #2: Lev Tarasov (lev@mun.ca)**
()

We thank Lev Tarasov for an excellent and detailed review and helpful comments. We learned a lot by working through his ideas and suggestions.

This part II submission of Albrecht et al examines a moderately sized ensemble of Antarctic glacial cycle runs with the PIK variant of the PISM ice sheet model. The ensemble runs are scored against a range of paleo and present−day (PD) constraints. The scored ensemble uses a reasonably state of the art model for ensemble glacial cycle contexts and adds to the literature of what various models will do for past AIS glacial cycles. I therefore see the study potentially worthy of publication in TC given the current bar. At some point in the future I hope that model−based studies will have the requisite level of uncertainty quantification to enable much more meaningful inferences about past ice sheet evolution. However even with the current bar, a number of significant deficiencies need to be addressed.

The experimental design has some significant problems that are not even discussed. The study only using 4 ensemble parameters. Briggs et al, (TC, 2013) for instance, have 12 ensemble parameters just for the climate forcing and 31 ensemble parameter in total. At least 1 of the 5 temperature related ensemble parameters in that study (Tmix1) was one of the most sensitive ensemble parameters (with generally more sensitivity to this than to the precipitation related parameter (PdeselevEXP) that best corresponds to the sole climate forcing parameter (PREC) in this submission. The lack of an ensemble parameter relevant to the temperature forcing is especially problematic given the stated context of providing a distribution of present−day ice sheet states for initializing future projection runs. That state will depend significantly in the 3D temperature field of the ice sheet, the uncertainty of which is not probed in this study. Ideally this would be remedied, but that would be a major endeavor. At the very least I expect a clear and complete discussion of model and experimental design weaknesses and associated relevance to given results. A summary of this should also be in the conclusions.

Again, we thank the reviewer and are glad that he considers the study in principle worthy for publication in The Cryosphere (TC). We understand that only 4 selected model parameter cannot map the whole multidimensional phase space of model states and that other independent parameters might be relevant, too. However, given the limited granted computational budget and the minimum in simulation length and resolution (see Sect. 2.2 in companion paper part 1 (Albrecht et al., 2019a)) we were able to run up to around 500 simulations. It is a compromise, but as we decided to use simple averaging instead of advanced statistical emulators that interpolate parameter space (Chang et al., 2016a,b), we were restricted to 4-5 most relevant parameters, in order to privide reasonable results for the ensemble (see Chang et al. (2014) for Greenland application), while more than 30 parameters (also with Latin HyperCube, as in Briggs et al., 2013) would require many thousand simulations to be sufficiently spaced. Briggs et al. (2014) compensated for their „low sample size of model runs relative to the dimension of the parameter space" with

„some emphasis ... on sensitivity to the choice of ensemble sieves.“

Regarding the PREC ensemble parameter in our study, this in fact does not correspond to the desert elevation effect coefficient PdeselevEXP in Briggs et al. (2013), as it only scales with the external temperature forcing, not with changes in the surface geometry. It would be more similar to PphaseEXP, if it would not scale with insolation but with temperature.

We also tested for different temperature forcings in the companion paper part 1 (Albrecht et al., 2019a), and found comparably little influence of the present-day temperature distribution (Sect. 3.1: parameterized or from model output, with or without PDD, different PPD paramters) and for different temperature forcings (Sect. 4.2) on the Antarcrtic Ice Sheet history with less than 1m SLE sensitivity for LGM and about 2m SLE for PD results. In the Briggs et al., 2013 study, Tmix1 showed infact a high variance of 10 mESL, but this was related not only to a present-day temperature parameterization, but also included insolation and sea-level forcing.

We have added a clearer discussion of model and experimental design deficiencies and associated relevance to given results to the revised manuscript.

Another major omission is a comparison of the ensemble results against the paleo data constraints. The chosen normalization of all score components against median scores means that the scores do not convey any information about absolute model fit to paleo data, only relative fit. It is therefore incumbent that the complete set of ensemble fits to paleo constraints be explicitly shown and discussed, eg as done in Briggs et al, 2014.

We added plots of individual paleo score fits as in Fig. 7-10 in Briggs et al, (2014) and respective discussion to the Supplementary Material B.

Furthermore, there are a number of claims and statements (detailed below) made that I find are indefensible, misrepresentative, and/or incorrect.

**Specific comments**

Large Ensemble of 256
−> Ensemble of 256
**with eg Briggs et al using a 2000 member ensemble, you can hardly call 256 a "Large Ensemble".**

With the label „large ensemble“ we here directly referred to the „LE“ definition of ensembles that Pollard et al. (2016) defined (they used 625 ensemble members),  „i.e., sets of hundreds of simulations over the last deglacial period with systematic variations of selected model parameters“. We provide a better quantitative classification of ensemble size in the revised manuscript:
*„In view of the even larger ensemble by Briggs et al. (2014) with 31 varied parameters and over 3,000 simulations, our ensemble with only four varied parameters and 256 simulations is of rather intermediate size, although we used a much finer model resolution.“*

"The model is calibrated against..."
−> The model is scored against...
**scoring a moderately sized ensemble is not calibration**

Ok, modified.

**what is the SSA enhancement factor value? It is never explicitly given in this study nor in the PART I of this submission.**

We agree that the SSA enhancement reference value of 0.6 is somewhat hidden in Sect. 2.3, Fig. 3 and Table. 1 of the companion paper (Albrecht et al., 2019a), where we find only little effect on LGM ice volume and almost no difference in deglacial or present-day ice volume, when values of 0.6 and 1.0 are compared. Our reference value agrees with the reference value in Briggs et al. (2014). For clarity we added a sentence to the parameter section 2.1:
*„In all ensemble runs we used for the SSA stress balance an enhancement factor of 0.6 (see Sect. 2.3 in companion paper) which is more relevant for ice stream and ice shelf regions."*

This analysis further constrains relevant model and boundary parameters by revealing clusters of best fit parameter combinations.
**Isn't that already previously stated in different words:**
"The model is calibrated[scored] against..."

We emphasize the new finding here by rephrasing:
*„This analysis reveals clusters of best fit parameter combinations and hence a likely range of relevant model and boundary parameters, rather than individual best fit parameters."*

Our Large Ensemble analysis also provides well−defined parametric uncertainty bounds and a probabilistic range of present−day states that can be used for PISM projections of future sea−level contributions from the Antarctic Ice Sheet.
**Kind of meaningless. I can think up a dozen metrics that would provide "well−defined parametric uncertainty bounds", each with different resultant ranges.**

We rephrased this sentence more generally as:
*„Our ensemble analysis also provides an estimate of parametric uncertainty bounds for the present-day state that can be used for PISM projections of future sea-level contributions from the Antarctic Ice Sheet."*

Nonconserving sub−glacial hydrology model
**as from review of part I: pretty crude to call this a model −>**
**Here we use the non−conserving sub−glacial hydrology parametrization**

We agree, the term „parameterization" would be more valid, but we actually refer to the non-conserving mode of the sub−glacial hydrology model, as cited in the previous sentence. We modified this in the manuscript as:
*„We use the non-conserving mode of sub-glacial hydrology model, which balances basal melt rate and constant drainage rate, to determine the effective pressure on the saturated till."*

**Missing brief (eg 1 sentence) description of bed thermal model.**

We have added more explanation to the part 1 companion paper and added also a sentence to the introduction of this study:

*„Geothermal heat flux based on airborne magnetic data from Martos et al., 2017 is applied to the lower boundary of a bedrock thermal layer of 2km thickness which accounts for storage effects of the upper lithosphere and hence estimates the heatflux at the ice-bedrock interface."*

Sub−shelf melting in PISM is calculated via PICO (Reese et al., 2018) from salinity and temperature in the lower ocean layers on the continental shelf (Schmidtko et al., 2014) in 18 separate basins based on (Zwally et al., 2015) adjacent to the ice shelves around the Antarctic continent
**the companion paper states that salinity was not varied:**
**"While salinity change over time in the deeper layers is neglected in this study"**
**and this should be made clear here.**

This is correct and we are sorry for this misunderstanding. We referred here to the observations of mean salinity and temperature in the lower ocean layers on the continental shelf by Schmidtko et al. (2014), to define the reference ocean state, while PICO in our study responds to changes in external ocean temperature forcing. We rephrased this paragraph as:

*„Sub-shelf melting in PISM is calculated via PICO (Reese et al., 2018) from observed salinity and temperature in the lower ocean layers on the continental shelf adjacent to the ice shelves around the Antarctic continent (Schmidtko et al., 2014) and as mean over 18 separate basins based on Zwally et al., 2015. PICO updates melt rates according to changes in ocean temperatures or the geometry of the ice shelves."*

use the Large Ensemble approach
**Why is this capitalized? "the" makes no sense as there**
**are lots of large ensemble approaches. Furthermore, has already**
**stated above, this is not a large ensemble.**

As mentioned above er here referred to the „LE" definition in Pollard et al., 2016. We make this clearer in the revised manuscript and avoid capital letters.

This method yields as reasonable results for an adequately resolved
parameter space as more advanced statistical techniques with means of
interpolating results between sparsely separated points in
multi−dimensional parameter space.
**I would strongly dispute this since full−factorial sampling**
**restricts one to a relatively small number of ensemble parameters.**
**The cited Pollard et al (2016) paper used an ensemble of**
**WAIS only simulations for the last 30 kyr. Ie all ensemble**
**members had identical initial conditions and identical**
**time evolving ice boundary conditions at the junction with**
**the East Antarctic ice sheet. This is a far cry from applying**
**4 ensemble parameters to the whole AIS for 2 glacial cycles.**

Pollard et al. (2016) only simulated WAIS, but Pollard et al. (2017) applied the same ensemble method to the whole Antarctic Ice Sheet, where also mantle viscosity profiles in a coupled GIA model have been varied. We want to emphasize that also our glacial cycle ensemble analysis has some focus on the last 30kyr and its impact on the present-day

state, also as most paleo constraints are limited to this period. Hence, the comparison to the latest PSU-ISM model results are not too far off. We added a sentence:

*„Yet, the full-factorial simple averaging method strongly limits the number of varied parameters for available computer resources such that only the most relevant parameters for each class of climatic and boundary conditions were pre-selected (in the companion paper) to cover a representative range of model responses.“*

It covers uncertainties within the Earth model for values of
1e19, 5e19, 25e19 and 100e19 Pa s.
**this study would benefit from better attention to relevant**
**litterature. While there is local support for viscosity as low as**
**1e19 Pa s on the Antarctic Peninsula, there is no support for even**
**5e19 over say the whole WAIS. Furthermore, the upper bound test**
**viscosity (and please use the more standard X10^21 units as**
**preferred in the GIA community) is half of the 95% "confidence"**
**upper bound of 2.0 X10^21 Pa s of Whitehouse et al, 2012 (GJI).**

We thank the reviewer for this comment, as recent literature often give the impression that Antarctic upper mantle viscosities have been overestimated previously. We apologize for using the wrong units in this paragraph, the covered range is actually 0.1-10.0 x 10^21 Pa s for the upper mantle viscosity, and hence much closer to the 95%-confidence range of 0.8-2.0 x10^21 Pa s in Whitehouse et al. (2012b) or the spatially-average of 0.2-1.0 x10^21 Pa s beneath whole Antarctica in Whitehouse et al. (2018), as cited in the companion paper. Units have been adjusted throughout the manuscripts.

This compilation also includes records of regional sea−level change (RSL), which has not been considered in this study since most of the sea−level signal is a result of the sea−level forcing with up to 140m rather than the model's ice dynamical response expressed in terms of sea−level equivalents, as PISM lacks a selfconsistent sea−level model
**Since the RSL data for Antarctica is above present−day elevation,**
**the above statement as written is incorrect. The RSL data is the**
**signal, and dominance of a far−field sea−level forcing would result**
**in sealevel below present−day.**

Yes, this was a badly formulated argument, we omitted the far-field sea level: *„This compilation also includes records of regional sea-level change above present-day elevation (RSL), which has not been considered in this study as PISM lacks a self-consistent sea-level model to account for regional self-gravitational effect of the order of up to several meters, which can be similar to the magnitude of post-glacial uplift.“*

We have actually tested for the addition of the RSL data type and found only little difference (slightly favoring higher viscosities) in the associated score pattern in parameter space among the ensemble members, when using the product of individual data type scores.

Mean−square−error misfit to observed grounding line location for the modeled Antarctic grounded mask (ice rises excluded) using a signed distance field
**I don't understand what you mean by signed distance as RMSE would**
**only care about unsigned distance. Or do you mean what we do in my**
**group: also track mean (ie not RMSE) error and use that to assign a**
**signed value to the RMSE?**

„Signed distance" is the name of a numerical technique for finding approximate solutions to the boundary value problems of the Eikonal equation using the fast marching method, here in two dimensions. The reviewer is right, that the differentiation between in and out (sign) is not considered in the mean square error in our study. We omit the term „signed" to avoid confusion: „*Mean-square-error misfit to observed grounding line location for the modeled Antarctic grounded mask (ice rises excluded) using a two-dimensional distance field approximation (https://pythonhosted.org/scikit-fmm).*"

5. UPL: Mean−square−error model misfit to modern GPS−based uplift rates on rock outcrops at 35 individual sites using the compilation by Whitehouse et al. (2012b, Table 2) including individual observational uncertainty
**Would be good to update the GPS data−set. Current GPS data versus**
**that approaching a decade old would make a significant difference in**
**observations and observational uncertainties.**

We thank the reviewer for this suggestion and we certainly consent that open data compilations should be updated and joined to serve the whole community. There are several groups with expertise in GPS processing, but according to Pippa Whitehouse (personal communication) there is no recent publication that documents GPS rates across the whole Antarctic continent, except for Schumacher et al. (2018). However, there are many different choices, which requires expert input and should fill a seperate publication. Also, one would need to bear in mind that simulation results of a coupled solid Earth model would be associated with the viscous dynamics, while the GPS signal also implies the elastic signal due to contemporary surface mass changes.
For this study we preferred to use similar datasets as in previous publications (i.e. Pollard et al., 2016, 2017) in order to have a better comparison between the individual model responses. But even for relatively large uncertainties in the older data, the data type UPL shows strong variations in individual ensemble scores with impact on the aggregated score accordingly.

Then the individual score $S_{i,j}$ is normalized according to the median to
**Why the mean versus the median?**

We follow closely the definition in Pollard et al. (2016; Sect. 2.4.1), which does not mean that we support all choices they did. The algebraic mean can be inappropriate if the values range over many orders of magnitude. However, in the 9 used datatypes we find similar values for mean and median (except for TROUGH mean, which is 34% larger), such that the effect on the total score is negligible (see Fig. R1). We used median for consistency reasons with Pollard et al. (2016). Also the RSL data type shows large difference between median and mean value, but this data type has not been considered in our score analysis.

[Figure]

*Fig. R1 Histogram of scores per data-type and median (in blue) and mean (green).*

As in Pollard et al. (2016) we also assume that each data type is of equal importance to the overall score, avoiding the inter−data−type weighting used by Briggs and Tarasov (2013); Briggs et al. (2014), which would favor data types of higher spatio−temporal density
**Would you still do this if say you only had 3 ELEV or EXT**
**datapoints? If all data were statistically independent, then one**
**would demand that data types of higher spatio−temporal density would**
**get more weight since in this case each and every datum should have**
**equal weight. You need to provide a better justification for this**
**choice then just blind citation of previous studies.**

The reviewer is definitely right, that data with small spatio-temporal influence should weight less. In fact, we have tested for inter-data weighting, similar to Briggs and Tarasov (2013), Briggs et al. (2014) and found only small influence on the overall pattern of the score distribution.

For an interdata-weight of PD(5):TOTUPL:TROUGH:ELEV:EXT of 0.5:0.05:0.15:0.2:0.1 we find that the best 25 unweighted scores (above 0.1 in green in Fig. R2a) also corresponds to the best weighted scores (below 0.75). This means that more than 200 simulations yield a relatively bad score in both definitions. In fact, there are 18 simulations with a weighted score below 0.75, which are below 0.1 in the unweighted case (blue). This distribution does not change much, when RSL is added as dataype with a low interdata-weight of 0.03 (Fig. R2b). In fact, most of those simulations would show similar scores if equal weights were attributed. So this is an effect of product vs. sum of individual scores.

[Figure]

*Fig. R2 Scatter plot of total scores in this study (best equals value 1.0) vs. an inter-data weighted score definition in Briggs & Tarasov (2013), with best value around 0.6 and larger scores meaning larger misfit. In the left-hand panel RSL datatype was added.*

We will also added supplement plots for individual data types as in Briggs et al. (2014), Fig. 5-10, Fig. R4 for RSL, Fig. R5 for ELEV and Fig. R6 for EXT:

[Figure]

*Fig. R3: RSL misfit as in Fig. 5-6 in Briggs et al., 2014, with same axis limits and data points (+ markers). Red dashed is mean of whole ensemble (no sieves applied), black is lower and blue upper relative sea-level history at site location, green dotted is best fit simulation. RSL was not used in the score analysis.*

[Figure]

*Fig. R4: ELEV misfit as in Fig. 7-9 in Briggs et al. (2014). Green is best fit simulation, blue is reconstruction at site location and date.*

[Figure]

*Fig. R5: EXT misfit as in Fig. 10 in Briggs et al. (2014). Green is best-fit simulation, red and magenta is reconstruction of grounding line extent.*

The parameter ESIA enhances the shear−dominated ice flow and hence ice thickness
**enhanced ice flow will not enhance (ie thicken) ice thickness but thin it**

That was a mistake: *„The parameter ESIA enhances the shear-dominated ice flow and hence yields ice thickening particularly in the interior of the ice sheet and therewith the total ice volume."*

For the upper range of mantle viscosities up to VISC = 1022 Pa s we find a normalized ensemble mean of 27% and 20%,
**This contradicts what you previously indicated was an upper bound value of 100 X10^19 Pas on page 4. ???**

This issued has been clarified above.

This value is also used in the GIA model ICE−6G (Peltier et al., 2015)
**kind of irrelevant since Peltier doesn't do regional tuning of Viscosity profiles.**

Has been omitted.

The best score ensemble members are found for intermediate mantle viscosities of VISC=5X10^20 Pa s and VISC=25X10^20 Pa s.
**THis again contradicts the values given on page 4. Furthermore,**
**25X10^20 Pa s is a high viscosity for the upper mantle (upper mantle is what**

**this half−space model best corresponds to)**

The unit question has been clairified above already. And yes, in the two-layer variant of the Lingle and Clark (1985) model, where the lower mantle represents one layer, without the low-viscosity channel beneath the lithosphere. As a half-space model this layer has indefinite thickness and one could think of the influence of the higher viscosity lower mantle, which is not explicitly considered here. We chose the parameter range large enough to find significant shifts in the scores, and therewith parameter values that can be excluded as a result of the analysis.

**3.2 Reconstructed sea−level histories**
**−> ice volume histories or sea−level contribution histories**

Yes, we preferred „*sea−level contribution histories"*.

the ensemble mean ice volume is 1.0m SLE below modern with a score−weighted standard deviation of around 2.7m SLE (volume of grounded ice above flotation in terms of global mean sea level equivalent as defined in Albrecht et al. (2019).
**Should compare this to published (paleo data−based) inferences of the Eemian high−stand as it makes it hard to fit current proxy−derived estimates for the Eemian high−stand given constraints on what Greenland could have contributed.**

This finding shows that the Antarctic Ice Sheet was somewhat smaller at Eemian. The indirect effect of Greenland melt is simply applied as sea-level forcing. Sutter et al., 2016 estimates around 3-4m SLE contribution of Antarctica during LIG, mainly trough WAIS collapse when a certain ocean temperature threshold is crossed. Also, the sea level high stand of the Eemian as a globally integrated signal suggests an Antarctic contribution of at least 1m ESL, and likley significant more (Cuffey and Marshall, 2000; Tarasov and Peltier, 2003; Kopp et al., 2009). This lower bound has been used as sieve criterion in Briggs et al. (2014). This additional information has been added to the manuscript.

The LGM ice volume increases for lower PPQ, lower PREC and lower ESIA values, while it seems to be rather insensitive to the choice of VISC
**All these relations would be expected as such.**

Added „*As expected, ...*", but we think it is good to remind the reader to this relations.

As MWP1a initiated the Antarctic Cold Reversal (ACR) with about two millenia of colder surface temperatures, a freshening of surface waters leading to a weakening of Southern Ocean overturning, resulting in reduced Antarctic Bottom Water formation, enhanced stratification and sea−ice expansion.
**This is not a sentence.**

This has been reformulated to:
„*The MWP1a initiated the Antarctic Cold Reversal (ACR), a period lasting for about two millenia with colder surface temperatures. This cooling induced a freshening of surface waters and lead to a weakening of Southern Ocean overturning, resulting in reduced Antarctic Bottom Water formation, enhanced stratification and sea-ice expansion."*

The modeled range between Last Glacial Maximum and present−day ice volume by Whitehouse et al. (2012a) is about 5.0X10^6 km3 (or 7.5 − 10.5m ESL, eustatic sea−level based on volume above flotation),...

**There is no point in listing all the exact ranges here and then showing those ranges in fig 11**
**Add the conversion factors to the figure key and have the paragraph focus on general comparison.**

We have re-formulated the whole paragraph and updated Fig. 11 accordingly.

Briggs et al. (2014) ... from PSU simulations for 40 km resolution
**To be accurate PSU + full visco−elastic isostatic adjustment bedrock**
**response with radially layered earth viscosity profile + different**
**subshelf melt, basal drag, climate forcing, and calving**
**treatments. So only PSU ice dynamics and thermodynamics.**

We have added more information on varied model parameters, ensemble size, resolution, simulation length and used Earth model to the revised manuscript.

Although the Large Ensemble method is limited to a comparably small number of values for each parameter
**no it is not. A full−factorial (grid) ensemble is limited. Not other**
**large ensemble approaches. And I don't understand why "Large**
**Ensemble" is capitalized. If you are choosing to equate a grid**
**ensemble as a "Large Ensemble", that makes no semantic sense. How are**
**readers supposed to differentiate between this useage and other**
**modelling studies that will use large ensembles under a different**
**sampling scheme? What about studies that have O(10) or more larger**
**ensembles?**

This is a good point. We assumed „large ensemble" to be a label for a class of ensembles that cover the whole (chosen) parameter phase space in contrast to sensitivity studies, in which parameter are varied separateley. We omited the term „large" in our studies and reformulated the paragraph as:
„*We have run an ensemble of 256 simulations over the last two glacial cycles and have applied a simple averaging method with full factorial sampling similar to* Pollard et al. (2016). *Although the this kind of ensemble method is limited to a comparably small number of values for each parameter...*"

**fig 11 plot and caption: there needs to be a note that some of the indicated studies have non symmetry distributions. Eg, Briggs et al, 2014 states : "likely between 5.6 and 14.3 m equivalent sea level (mESL), and with less confidence >10 mESL" = 4.0 X 10^6 km^3 of ice. The whitehouseBently12b datapoint is also problematic as that GJI paper never discusses ice volume or total sea level changes. The whitehouseBently12a does (and is cited in the preceeding discussion) and provides an uncertainty range but the plot only shows a single datapoint with no uncertainty range**

OK, we added:
„*The provided uncertainty ranges are not necessarily symmetric, e.g. the upper range in Briggs et al. (2014) has less confidence than the lower range.*" to the figure caption.

Regarding the datapoint in the Whitehouse et al. (2012a) study, we have contacted Pippa Whitehouse and she confirmed that the given range is the total range of simulated ice volumes rather than a standard deviation. The single plotted data point is the best fit

simulations and located at the lower end of that range. We have added information on this in the revised manuscript.

**fig 12 please increase the font size of the colour key for those of**
**use with ageing eyes...**

We increased the fontsize in Fig. 12-14.

**fig 14: please use a higher contrast colour scale, to make the**
**comparison more discernable. Even better would be the addition**
**of a difference plot to make clear what the differences are.**

We actually tested different color schemes for Fig. 14, and we agree that spectral colormaps may better cover the full range of surface velocity over several orders of magnitude. However, we want to emphasize here that the general arterial pattern of ice streams reaching far into the inland ice sheet is reasonable well reproduced, and preferred this seqential colormap with of model and observations side by side. An anomaly or (root-square-error) plot can be somewhat misleading, as confined ice streams may be slightly shifted in location or ice shelf velocity. In fact, the velocity mismatch is part of the scoring scheme and it can help to identify regions of under- or overestimation, such that we added a difference plot as suggested by the reviewer and increased the contrast and the range of the colormaps.

**fig 15−17 are hardly mentioned in the main text, with no consideration**
**of details. Eg, figure 17 has 7 timeseries, not one of which is individually**
**refered to in the text. So why is this figure in the paper?**

Those figures are made for comparison with a previous study on MWP-1A (Golledge et al., 2014) and have been referenced only once in the text. We added many more information on the distinct deglacial and regrowth phases with figure details to the manuscript.

3.4 Comparison to previous large ensemble study
**−> Comparison to Pollard et al. (2016) ensemble study**
**Your current title implies there was only one large ensemble**
**and the subsequent text implies it Pollard et al. (2016).**
**Briggs et al was a much larger ensemble study...**

We have drawn a better concerted picture in the revised manuscript.

while other parameters that affect the modern grounded ice areas are sufficiently
constrained by earlier studies
**This is not a fact, at best state they claim this.**

Changed.

In their ensemble analysis Pollard et al. (2016) included an iceberg calving par
ameter. Our 'eigencalving' model provides a fair representation of calving front dynamics
independent of the climate conditions (Levermann et al., 2012).
**What does "fair" mean? Be precise.**

Changed to: „Our 'eigencalving' parameterization provides a representation of calving front dynamics, which in first order yields present-day calving front positions (Levermann et al.,

*2012). This paramterization is rather independent of the climate conditions, variations of the 'eigencalving' parameter show only little effect on sea-level relevant ice volume (see companion paper Albrecht et al. (2019a))."*

As we used the PICO model (Reese et al., 2018) that includes physics to adequately represent melting and refreezing also for colder than−present climates, we have chosen other parameters in our ensemble,
**What about the large uncertainty is subshelf ocean temperature? Above you invo**
ke this
**to explain the lack of any MWP1a signal compared to eg Golledge et al, 2014**

We have actually considered the effect of intermediate ocean warming during ACR on the PICO sub-shelf melt rates in Sect 5.2 in the companion paper. This reference and some more details have been added to the revised manuscript. The Golledge et al. (2014) study used a rather crude estimate of sub-shelf melt rates from a scaling between LGM and modern states, which yields extremely high melt rates of up to 100m/yr for present climate, without considering the overturning circulation in the ice shelf cavity nor boundary effects between ice and ocean.

comparably small mantle viscosity around VISC = 5−25X10^20 Pa s,
**small compared to what? I wouldn't call these small upper Mantle viscosities f**
or Antarctica

We were actually talking about the lower tested range and omitted „comparably small".

Due to the comparably coarse resolution and the high uncertainty that comes with the strong non−linearity (sensitivity) of the system we here discuss rather general patterns of ice sheet histories than exact numbers.
**This non−linearity is another reason to increase the number of ensemble parame**
ters.

We added: „..., which would require a larger ensemble with an extended number of varied parameters."

Our ensemble−mean lies at the upper range of most previous studies, except for the large ensemble study by Pollard et al. (2017) with only 3−8m SLE since LGM, as their score algorithm favored the more rigid and hence thinner ice sheet configurations.
**incorrect in that half of your stated range is below the favoured**
**range of Briggs et al (2014) who state**
"The LGM ice volume excess relative to present−day is likely between 5.6 and 14.
3 m
equivalent sea level (mESL), and with less confidence >10 mESL
**Furthermore, your sentence contradicts itself as currently written.**

Right, this has been formulated rather crudely. Converted into total ice volume, our score-weighted range of 5.8+-2.0 mio. km3 relative to present overlaps with the less confidence upper range (>4.0 mio. km3) of Briggs et al. (2014) of with 2.2-5.7 mio km3, while there is almost no overlap to the range found by Pollard et al. (2017) of 3.4+-0.7 mio. km3 relative to present (the 3-8m are associated with the approximate range of best fit ensemble members in their Fig. 2, as discussed with Dave Pollard). Their range is quite consistent

with their previous study on WAIS only with 3.2+-1.6 mio. km3 (Pollard et al., 2016). Also Golledge et al., (2012, 2013) are below our range with 2.7 and 3.4 mio km3 respectively, while in contrast the value of 5.8 mio km3 in Golledge et al. (2014) (relative to Bedmap2) is close to our ensemble mean.

We rephrased this paragraph (without the numbers) accordingly: „*Our ensemble-mean ice volume anomaly between LGM and present is close to the best fit value found in Golledge et al. (2014) and Whitehouse et al. (2012a), while the ANICE best fit values (Maris et al., 2014, 2015) lie in the lower uncertainty range of our study. In contrast, the large PSU-ISM ensemble mean by Pollard et al. (2016, 2017) as well as the high confidence (lower) range in Briggs and Tarasov (2013) are found below the uncertainty range of our study. Also the PISM equilibrium values by (Golledge et al., 2012, 2013) are clearly below the uncertainty range of our ensemble.*"

Previous studies with PISM Golledge et al. (2014) suggest
**english and punctuation...**

„*A previous PISM study suggest that the oceanic forcing at intermediate levels can be of opposite sign as compare to the surface forcing, as likely happened during the two millennia of Antarctic Cold Reversal following the MWP1a, causing earlier and larger sea-level contributions from Antarctica (Golledge et al., 2014).*"

In this study we used the
Bedmap2 topography remapped to 16 km resolution without local adjustments
**Does this actually belong in the "Conclusions"?**

This sentence is an explains the different rebound behaviour with respect to the previous study, we switched the order to: „*In contrast to Kingslake et al. (2018), we used the remapped topography without local adjustments, such that in only about 20% of the score-weighted simulations this region re-grounded.*"

provides model and observation calibrated parameter constraints for
projections of Antarctic sea−level contributions
**awkward and somewhat indecipherable. Do you just mean**
**"data−contrained projections of .. using PISM"?**

We are sorry for using the terms „calibrated" and „constrained" as synonyms, we corrected for this misunderstanding troughout the manuscript.

With the best−fit simulation parameters we have participated in the initMIP−Antarctica model intercomparison (Seroussi et al., 2019, PISMPAL3).
**This is not a conclusion**

Ok, has been moved to results section and caption of Table 1.

**appendix A : is referred two a couple of times, but without any**
**statement of what the takeaway from the appendix is.**

„*One key parameter for the onset of retreat could be the minimal till friction angle on the continental shelf with values possibly below 1.0. More discussion of the interference of*

*basal parameters in terms of an additional ensemble analysis is given in Supplementary Material A."*

*„Friction underneath the modern ice shelves is highly relevant, in particular during the deglaciation, as we have discussed in the companion paper Albrecht et al. (2019a). However, instead of choosing the friction coefficient underneath the modern ice shelves as ensemble parameter (Pollard et al., 2016, 2017) we decided on the sliding exponent as uncertain parameter for the entire Antarctic Ice Sheet. In fact we have run an additional ensemble analysis for four basal sliding and hydrology parameters only, including friction underneath modern ice shelves and discussed the results in the Supplementary Material A. As the main deglacial retreat (in the basal ensemble mean and in the best fit simulation therein) occurs a few thousand years earlier (closer to MWP-1A) the corresponding scores are even better than for the best fit simulation of the base ensemble (for same sliding exponent but smaller minimal till friction angle)"*

In the basal sub−ensemble we find even better scores than for the best fit param eter combination in the large ensemble (here no. 8102, see Fig. A.
...
However, best fit to the nine constraints are found for
the basal ensemble..
which agrees with the best fit values of large ensemble.
**The above two statements contradict each other**

Sorry for this ambiguity, we were actually talking about the best fit parameter values and not the scores:
*„However, best fit to the nine data constraints are found for the basal ensemble in the middle range of PPQ = 0.5–0.75 and the lower range of till water decay rates of TWDR = 0.5-1 mm/yr (1.55–3.1×10−11 m/s), which agrees with the best fit parameter combination of the base ensemble (PPQ=0.75 and TWDR=1 mm/yr). „*

[revised manuscript text omitted]
  base ensemble (here no. 8102, see Fig. S1)  that covers also climatic, Earth and ice-internal parameters. Best scores are found in particular for smaller minimal till friction angles PHIMIN = 0.5–1°, but also for rather high values of the fraction of the effective overburden pressure at which excess water drains, here FEOP = 4–16%. These values are higher than those used in the  base ensemble. However, best fit to the nine data constraints are found for the basal ensemble in the middle range of PPQ = 0.5–0.75 and the lower range of till water decay rates of TWDR = 0.5-1 mm/yr ($1.55$–$3.1 \times 10^{-11}$ m/s), which agrees with the best fit  parameter combination of the base ensemble (PPQ=0.75 and TWDR=1 mm/yr). The LGM volume of the best fit simulation of the basal ensemble is similar to the best fit simulation of the  base ensemble (cf. Figs. S2 and 15), however deglacial retreat occurs a few thousand years earlier for lower PHIMIN.

[Figure]

Figure S2: Snapshots of grounded ice thickness anomaly to present-day observations (Bedmap2; Fretwell et al., 2013) over the last 15 kyr for best-fit simulation in the basal ensemble. At LGM state grounded ice extends towards the edge of the continental shelf, with much thicker ice than present mainly in West Antarctica. Retreat of the ice sheet initiates between 12 and 11 kyr BP and halts already latest 8 kyr in all large ice shelf basins of Ross, Weddell Sea, Amery and Amundsen Sea. East Antarctic Ice Sheet thickness is underestimated throughout the deglaciation period (light blue). Compare Fig. 2 in Golledge et al. (2014).

[Figure]

Figure S3: During deglaciation the score-weighted ensemble mean (green) shows most of the sea-level change rates between 14.5 kyr BP (MWP1a) and 8 kyr BP with mean rates around $1\,\mathrm{mm\,yr^{-1}}$, while the best-score simulation (red) reveals rates of sea-level rise of up to $4\,\mathrm{mm\,yr^{-1}}$ (100 yr bins) in the same period (cf. Golledge et al., 2014, Fig. 3 d). In contrast to the  base ensemble (cf. Fig. 9c) the basal ensemble shows a much earlier deglacial retreat and no regrowth during the late Holocene.

~~Development of PISM is supported by NASA grant NNX17AG65G and NSF grants PLR-1603799 and PLR-1644277.The authors gratefully acknowledge the European Regional Development Fund (ERDF) , the German Federal Ministry of Education and Research and the Land Brandenburg for supporting this project by providing resources on the high performance computer system at the Potsdam Institute for Climate Impact Research. Computer resources for this project have been also provided by the Gauss Centre for Supercomputing/Leibniz Supercomputing Centre (www.lrz. de)~~

1025

under Project-ID pr94ga and pn69ru.T. A. is supported by the Deutsche Forschungsgemeinschaft (DFG)in the framework of the priority program "Antarctic Research with comparative investigations in Arctic ice areas" by grant LE1448/6-1 and LE1448/7-1. We thank Dave Pollard for sharing ensembleanalysis scripts and for valueable discussions.

**Supplementary Material B: Misfit to individual paleo data types**

This appendix compares model results with corresponding geological data types (AntICEdat from Briggs and Tarasov (2013) ) used in the ensemble scoring. This absolute misfit is important information as all scores are normalized against their median (relative fit) in order to calculate the aggregated scores. Thereby, we want to demonstrate how well the ensemble simulations span the data constraints and hence potentially represent reasonably realistic ice-sheet behavior.

Fig. S4 compares elevation vs. age for all 256 runs with cosmogenic data at 26 sites (ELEV; Briggs and Tarasov, 2013) with a median age of constraint of 9.6 kyr. We find a good fit in parts of East Antarctica (e.g. Framnes Mts. (1201-1203)) and in parts of the Ross sector (e.g. Clark Mts. (1405), Allegheny Mts. (1406) or Eastern Fosdick Mts. (1408)), while in the West Antarctic Ice Sheet there is quite a large spread among the ensemble misfit of up to 1,000 m in surface elevation, with ensemble mean misfits of up to 1,000 m. This is due to the fact that in many ensemble simulations the large ice shelves of Ronne-Filchner, Ross and Amery do not become afloat in time, while the best-fit simulation (green markers) shows quite a good fit, although some regions remain thicker than observed until present (Fig. 12).

Fig. S5 shows the misfit of simulated grounding lines retreat for all ensemble simulations at 27 marine core sites (EXT; Briggs and Tarasov, 2013) , which are relatively well distributed around the Antarcric Ice Sheet with a median age of 16.6 kyr, the oldest data point 30.7 kyr. Generally, simulated grounding line retreat occurs later than in most of the observations, less than 5 kyr near Victoria Land, Ross Sea and along the Antarctic Peninsula (2303, 2402-2403, 2602-2608) and less than 10 kyr in the Amundsen Sea, and Weddell Sea (2502, 2609, 2701). At some locations (Dronning Maud-Enderby Land (2101-2103) or at Victoria Land (2304)), however, the ensemble never reproduces the recorded open ocean conditions or grounding line retreat event, respectively.

Although not used as constraint in our scoring scheme, Fig. S7 shows the misfit of modelled relative sea level in all ensemble simulations with respect to 96 RSL proxy records at eight sites (RSL; Briggs and Tarasov, 2013) , with a median age of 5.0 kyr. The data for each site fall well within the overall model envelope (upper and lower bound indicated) with best fits at Syowa Coast (9101), Larsemann Hills (9201), Vestfold Hills (9202), Windmill Islands (9301), and Marguerite Bay (9601) and King George Island (9602), while in Victoria Land the model ensemble generally overestimates regional sea level (Terra Nova Bay (9401) and Southern Scott Coast (9402)).

From each data type misfit we obtain a ensemble distribution of misfits (Fig. S6), which can be rather normal (e.g. for EXT), exponential (e.g. TOTUPL) or long-tail (e.g. TOTDH). In order to calculate aggregated scores we normalize by the median value, which yields for most data types similar results as the mean value, except for TROUGH (34% difference). The corresponding variability of each of the resultant normalized scores hence contribute different skills to the aggregated score. Generally, grounding-line related (TOTE, TOTGL, THROUGH) and ice volume-related data-types (TOTDH) show similar individual score patterns (not shown here) with ensemble standard deviations of 0.1-0.2. In the aggregated score this patterns becomes even more pronounced, while paleo scores (ELEV and EXT) and ice shelf extent (TOTI) show only little variation (<0.1) among the ensemble, and hence only little effect in the aggregate score pattern.

[Figure]

Figure S4: ELEV observations (colored diamonds, dark and light blue indicate last 10 kyr or 20-10 kyr observational interval) taken from database by Briggs and Tarasov (2013) , ensemble results (black circles), upper and lower bounds from base ensemble (red triangles), and computed misfits (lower panel) for different Antarctic Peninsula sectors, indicated by vertically dashed lines and labels between panels. Green dots correspond to best-fit simulation. Compare to Fig. 7–9 for in Briggs et al. (2014) with same data-point identifiers.

[Figure]

Figure S5: EXT observations and ensemble results as in Fig. 10 in Briggs et al. (2014) . Black circles represent the 256 ensemble simulations with the best-fit simulation in green. Red indicate the grounding line retreat (GLR) two-way constraint types, magenta the open marine conditions (OMC) one-way constraint types. Dashed horizontal lines and associated labels segregate and identify the different sectors.

[Figure]

Figure S6: Histogram of misfits per data-type with median (in blue) and mean (green).

[Figure]

Figure S7: Regional sea level (RSL) data points and ensemble sea level curves for the 8 data sites, analogous to Fig. 5–6 in Briggs et al. (2014) , upper panels in EAIS, lower panels in Antarctic Peninsula and Ross sector. Observed RSL data points are colour coded according to the constraint they provide: two-way (light blue, dated past sea level); one-way lower-bounding (mauve, past sea level above or maximum age of beach) or one-way upper-bounding (orange, past sea level below or minimum age beach). For a detailed description of the RSL datasets and its processing, refer to Briggs and Tarasov (2013) . RSL has not been used as constraint in this study.

---

## Author Response (AR3)

Response to Editor Decison by Alexander Robinson (16 January 2020)

Find in **blue** the referee's comments and in **black** the author's response.

**Editor Decision: Publish subject to technical corrections**
Comments to the Author:
Dear authors,

Your revision is almost ready for publication. See only the minor additional changes to be made below:

Title: Glacial cycles simulation => Glacial cycle simulations [As with the companion paper]

L570: assistence => assistance

Best regards,
Alex

Thanks for pointing out this grammatical error in the title adn in the acknowledgements, we corrected this accordingly.
* * *
Response to Editor Decison by Alexander Robinson (04 December 2019)

We would like to thank the editor for his positive assessment and the careful read identifying and reporting the many small typos. We hope that the revised manuscript is in much better shape now.

Comments to the Author:
Dear Authors,

I find only minor improvements necessary in the manuscript before publication. However, many small typographical errors were present. I have found some of them, but I would recommend another careful reading. Also note, that I did not check the SI for English. I look forward to your submission of a revised version.

Best regards,
Alex

L81: simulations => simulation

ok

L81: between PISM => between the PISM

ok, we used „a PISM simulation"

L99: Ensemble parameter => Ensemble parameters

ok

L102: Confusing: "The selected parameters passed the two main criteria and …". What are the two main criteria, have they been defined already? Or perhaps is the "and" a typo?

„The selected parameters passed the two **following** main criteria **of** (1) **showing** … (2)..."

L174: location => locations

ok

L179: to model grid => to the model grid

ok

Fig. 7: I believe the color-legend should be unitless, not in "[m]".

correct!

L379: during ACR period => during the ACR period

ok

L416: parameter => parameter values

ok

L429: Typo "estinate fo"

ok

L432: really => very

ok

L466: At last glacial maximum => At the Last Glacial Maximum [Should this be "During the deglaciation"?]

Right, we joinded the 10 kyr, where deglaciation starts.

L470-471: Same comment as above.

We joinded the first and last sentence: „At the Last Glacial Maximum **around 15 kyr BP** the sea-level relevant volume history of the best-score simulation is close to the ensemble mean (Fig. 9)**, which agrees well with reconstructions by the RAISED Consortium (Bentley et al., 2014, cf. Fig. 7 a)."**

L478-481: Confusing as written, please revise.

„Maximum ice volume change rates are found accordingly in the period **between 10 and 8 kyr BP** with **on average** -1.4 mm/yr SLE (or -660 Gt/yr) and in the period **between 8 and 6 kyr BP** with **on average** -2.4 mm/yr SLE (or -1,300 Gt/yr, Fig. 16). **In the 100 yr running mean of the ice volume change rate we find a** peak of around -5 mm/yr SLE at 7.5 kyr BP (or -3,300 Gt/yr, compare black and khaki line in Fig. 17). This rate of change is significantly larger than in the ensemble mean **with up to -2 mm/yr SLE, as the mean retreat becomes smoothed over a longer deglacial period** (see Fig. 9 c).“

L485: Discharge and melting should be positive quantities here, I believe.

ok

L499-500: Discharge and melting should be positive quantities here, I believe.

ok

Fig. 16: It would be more intuitive for the panel year labels to be chronological (e.g., 10-8 kyr BP instead of 8-10 kyr BP).

We agree, the labels were inspired by Fig. 4 in Golledge et al., 2014 as they not only characterize a period but also the anomaly calculation (subtraction). We modified the plot as suggested.

L510: parameters => parameter

ok

L511: whole => all of

ok

L512-513: Use of semi-colon is confusing here. Perhaps an "and"?

ok

L518: for instance climatic forcing => climate forcing, for instance,

ok

L527: a erosion => an erosion

ok

L533: Remove parentheses around citation.

ok

L545: small ESIA => small ESIA values

ok

L557: As => As the

ok

L573: which in first order => which, to the first order,

ok

L580: Typo ". the"

ok

L582: One could argue that the scaling of ocean temperatures to atmospheric temperatures (of ~0.4) could be quite uncertain. Perhaps you could add a sentence here as to why you did not choose to vary this parameter?

Oh yes, this is important and has been discussed in the sensitivity paper by scaling the ocean temperature forcing by 60%, which corresponds to a total scaling of ocean temperatures with surface temperature by about 25%, associated with an earlier warming signal that could potentially contribute to the initiation of the deglaciation. We added: „Also for variation of the scaling constant of ocean input temperatures with surface temperature the glacial ice volume showed a comparably low sensitivity (see Sect. 4.3 in the companion paper Albrecht et al. (2019))."

L589: rather wide => to be rather broad

ok

L610: reconstruction => reconstructions

ok

L621: can not => cannot

ok

L631: Strongest => The strongest

ok

L632: most pronounced => the most pronounced

ok

L632: Grounding => The grounding

ok

L656: grounding line retreat => grounding-line retreat

ok

L666: a extensive => an extensive

ok

L668: The here presented paleo simulation ensemble analysis => The paleo-simulation ensemble analysis presented here

ok

L668: "with PISM" doesn't fit well, please rephrase.

[revised manuscript text omitted]
 $\underline{0.5 \, \text{mm/yr} \, \text{mm/yr}}$ $(1.55 \times 10^{-11} \, \text{m/s} \, \text{m/s})$, 1 $\text{mm/yr}$

- FEOP: For this fraction of the effective overburden pressure (for details see Bueler and van Pelt, 2015, Sect. 3.2), excess water will be drained into a transport system in the case of saturated till. Sampled values are $\underline{1\%, 2\%, 4\%. 8\%}$ and $\underline{32\%}$.

- PPQ: as in the ensemble (see Sect. 3.1)

[Figure]

Figure S1: Aggregated score for 318 ensemble members (4 model parameters, 4-5 values each) showing the distribution of the scores over the full range of plausible basal parameter values. The score values are computed versus geologic and modern data sets, normalized by the best score in the ensemble, and range from <0.01 (bright yellow, no skill) to 1 (dark red, best score) (cfs. Pollard et al., 2016, Figs. 2 + C1), on a logarithmic color scale. The four parameters are the effective overburden pressure fraction FEOP (outer y-axis), the minimal till friction angle on the continental shelf PHIMIN (outer x-axis), the tillwater decay rate TWDR (inner y-axis) and the power-law sliding pseudoplasticity exponent PPQ (inner x-axis). In the lowest row, only four ensemble scores are shown for 32% of effective overburden pressure fraction, just to ascertain that aggregated scores decline for larger FEOP.

965     In the basal sub-ensemble we find even better scores than for the best fit parameter combination in the base ensemble (here no. 8102, see Fig. S1) that covers also climatic, Earth and ice-internal parameters. Best scores are found in particular for smaller minimal till friction angles PHIMIN = 0.5–1°, but also for rather high values of the fraction of the effective overburden pressure at which excess water drains, here FEOP = 4–16%. These values are higher than those used in the base ensemble. However, best fit to the nine data constraints are found for the basal ensemble in the middle range of PPQ = 0.5–0.75 and the lower range of till water decay rates of TWDR = 0.5-1  mm/yr (1.55–3.1×10$^{-11}$  m/s), which agrees with the best fit parameter combination of the base ensemble (PPQ=0.75 and TWDR=1  mm/yr). The LGM volume of the best fit simulation of the basal ensemble is  considerably smaller (4.5 m SLE) than in the best fit simulation of the base ensemble (cf. Figs. S2 and 15),  and deglacial retreat occurs a few thousand years earlier for lower PHIMIN ( Fig. S3).

[Figure]

Figure S2: Snapshots of grounded ice thickness anomaly to present-day observations (Bedmap2; Fretwell et al., 2013) over the last 15  kyr for best-fit simulation in the basal ensemble. At LGM state grounded ice extends towards the edge of the continental shelf, with much thicker ice than present mainly in West Antarctica. Retreat of the ice sheet initiates between 12 and 11  kyr BP and halts already latest 8  kyr in all large ice shelf basins of Ross, Weddell Sea, Amery and Amundsen Sea. East Antarctic Ice Sheet thickness is underestimated throughout the deglaciation period (light blue). Compare Fig. 2 in Golledge et al. (2014).

[Figure]

Figure S3: During deglaciation the score-weighted ensemble mean (green) shows most of the sea-level change rates between 14.5  kyr BP (MWP1a) and 8  kyr BP with mean rates around 1  mm yr$^{-1}$, while the best-score simulation (red) reveals rates of sea-level rise of up to 4  mm yr$^{-1}$ (100  yr bins) in the same period (cf. Golledge et al., 2014, Fig. 3 d). In contrast to the base ensemble (cf. Fig. 9c) the basal ensemble shows a much earlier deglacial retreat and no regrowth during the late Holocene.

**Supplementary Material B: Misfit to individual paleo data types**

This  supplement compares model results with corresponding geological data types (AntICEdat from Briggs and Tarasov (2013)) used in the ensemble scoring. This absolute misfit is important information as all
980    scores are normalized against their median (relative fit) in order to calculate the aggregated scores. Thereby, we want to demonstrate how well the ensemble simulations span the data constraints and hence potentially represent reasonably realistic ice-sheet behavior.

     Fig. S4 compares elevation vs. age for all 256 runs with cosmogenic data at 26 sites (ELEV; Briggs and Tarasov, 2013) with a median age of constraint of 9.6  kyr. We find a good fit in parts of East Antarctica
985    (e.g. Framnes Mts. (1201-1203)) and in parts of the Ross sector (e.g. Clark Mts. (1405), Allegheny Mts. (1406) or Eastern Fosdick Mts. (1408)), while in the West Antarctic Ice Sheet there is quite a large spread among the ensemble misfit of up to 1,000  m in surface elevation, with ensemble mean misfits of up to 1,000  m. This is due to the fact that in many ensemble simulations the large ice shelves of Ronne-Filchner, Ross and Amery do not become afloat in time, while the best-fit simulation (green markers) shows quite a good fit, although some
990    regions remain thicker than observed until present (Fig. 12).

     Fig. S5 shows the misfit of simulated grounding lines retreat for all ensemble simulations at 27 marine core sites (EXT; Briggs and Tarasov, 2013), which are relatively well distributed around the Antarcric Ice Sheet with a median age of 16.6  kyr, the oldest data point 30.7  kyr. Generally, simulated  grounding-line retreat occurs later than in most of the observations, less than 5  kyr near Victoria Land, Ross
995    Sea and along the Antarctic Peninsula (2303, 2402-2403, 2602-2608) and less than 10  kyr in the Amundsen Sea, and Weddell Sea (2502, 2609, 2701). At some locations (Dronning Maud-Enderby Land (2101-2103)

or at Victoria Land (2304)), however, the ensemble never reproduces the recorded open ocean conditions or  grounding-line retreat event, respectively.

Although not used as constraint in our scoring scheme, Fig. S7 shows the misfit of modelled relative sea level in all ensemble simulations with respect to 96 RSL proxy records at eight sites (RSL; Briggs and Tarasov, 2013), with a median age of 5.0 kyr. The data for each site fall well within the overall model envelope (upper and lower bound indicated) with best fits at Syowa Coast (9101), Larsemann Hills (9201), Vestfold Hills (9202), Windmill Islands (9301), and Marguerite Bay (9601) and King George Island (9602), while in Victoria Land the model ensemble generally overestimates regional sea level (Terra Nova Bay (9401) and Southern Scott Coast (9402)).

From each data type misfit we obtain a ensemble distribution of misfits (Fig. S6), which can be rather normal (e.g. for EXT), exponential (e.g. TOTUPL) or long-tail (e.g. TOTDH). In order to calculate aggregated scores we normalize by the median value, which yields for most data types similar results as the mean value, except for TROUGH (34% difference). The corresponding variability of each of the resultant normalized scores hence contribute different skills to the aggregated score. Generally, grounding-line related (TOTE, TOTGL, THROUGH) and ice volume-related data-types (TOTDH) show similar individual score patterns (not shown here) with ensemble standard deviations of 0.1-0.2. In the aggregated score this patterns becomes even more pronounced, while paleo scores (ELEV and EXT) and ice shelf extent (TOTI) show only little variation (<0.1) among the ensemble, and hence only little effect in the aggregate score pattern.

[Figure]

Figure S4: ELEV observations (colored diamonds, dark and light blue indicate last 10  kyr or 20-10  kyr observational interval) taken from database by Briggs and Tarasov (2013), ensemble results (black circles), upper and lower bounds from base ensemble (red triangles), and computed misfits (lower panel) for different Antarctic Peninsula sectors, indicated by vertically dashed lines and labels between panels. Green dots correspond to best-fit simulation. Compare to Fig. 7–9 for in Briggs et al. (2014) with same data-point identifiers.

[Figure]

Figure S5: EXT observations and ensemble results as in Fig. 10 in Briggs et al. (2014). Black circles represent the 256 ensemble simulations with the best-fit simulation in green. Red indicate the  grounding-line retreat (GLR) two-way constraint types, magenta the open marine conditions (OMC) one-way constraint types. Dashed horizontal lines and associated labels segregate and identify the different sectors.

[Figure]

Figure S6: Histogram of misfits per data-type with median (in blue) and mean (green).

[Figure]

Figure S7: Regional sea level (RSL) data points and ensemble sea level curves for the 8 data sites, analogous to Fig. 5–6 in Briggs et al. (2014), upper panels in EAIS, lower panels in Antarctic Peninsula and Ross sector. Observed RSL data points are colour coded according to the constraint they provide: two-way (light blue, dated past sea level); one-way lower-bounding (mauve, past sea level above or maximum age of beach) or one-way upper-bounding (orange, past sea level below or minimum age beach). For a detailed description of the RSL datasets and its processing, refer to Briggs and Tarasov (2013). RSL has not been used as constraint in this study.